# Trigonometric gradient microstructures in additively manufactured single crystals enable strength-ductility synergy and programmable performance

Zixu Guo[1,5], Yang Li[2,5], Lei Fan[1], Shiwei Wu[1], Daijun Hu [1], Guochen Peng[1], Feng Lin [2], Yong-Wei Zhang [3], Yilun Xu [3,4] ✉ & Wentao Yan [1] ✉

Additively manufactured (AM) single crystals (SXs) show great promise for extreme-environment applications. AM process enhances gradient microstructures around dendrites, including dislocation densities, matrix channel width, precipitate area, and elemental concentrations. Here, we leverage a unified trigonometric function describing all gradient microstructures in AM SXs, to quantify their effects and enable programmable performance. We reveal that trigonometric gradient microstructures (TGMs) can overcome strength-ductility trade-off, particularly at elevated temperatures. In contrast, conventional gradient microstructures requiring post-treatment improve strength at the expense of ductility. This benefit is attributed to the superposition relationship between initial density-graded dislocations and other TGMs, rather than geometrically necessary dislocations in conventional understanding. High-throughput simulations reveal linear correlations between TGM intensity and mechanical properties. By mapping performance against TGMs, we can tailor strength and elongation by tuning TGMs. This study deepens the understanding of gradient microstructures around columnar dendrites in AM alloys and provides guidance for tailoring mechanical properties.

Eliminating grain boundaries to produce single-crystal (SX) alloys, is crucial for critical applications in extreme environments, including elevated temperatures[1], corrosion[2], and oxidation[3]. For instance, compared with polycrystals, SX alloys improve the maximum allowable temperature by 150 K in Ni-based alloys[4,5], and 200 K in TiAl intermetallic[6]. Hence, SX alloys are in urgent demand for some high-end aerospace applications, such as turbine blades in advanced aero-engines[4,5]. Additive manufacturing (AM), as one of the emerging technologies, is promising to achieve flexible near-net-shaping of SX

parts with low porosity, superior performance, and tunable properties[7-10]. In addition, the AM process is characterized by an extremely high cooling rate and a large temperature gradient. Taking AM Ni alloys as an example, these two indices exceed $2 \times 10^5$ K/s and $5 \times 10^5$ K/m[11,12], which are nearly $10^5$ and 200 times higher than those in casting[13]. These two distinctive characteristics could significantly intensify gradient microstructures around dendrites in SXs during rapid solidification, showing the potential to yield superior mechanical properties.

[1]Department of Mechanical Engineering, National University of Singapore, Singapore, Singapore. [2]Department of Mechanical Engineering, Tsinghua University, Beijing, China. [3]Institute of High Performance Computing (IHPC), Agency for Science, Technology and Research (A*STAR), Singapore, Singapore. [4]Department of Materials, Imperial College, London, UK. [5]These authors contributed equally: Zixu Guo, Yang Li. ✉e-mail: yilun.xu@imperial.ac.uk; mpeyanw@nus.edu.sg

Geometrically necessary dislocations (GNDs) induced by strain gradients, serve as an additional source of strengthening in various metallic systems[14–17]. However, the AM SXs with strong initial density-graded dislocations and residual stresses[18], potentially challenge the conventional GND-dominated strengthening mechanism related to gradient microstructures[19,20]. Besides, in as-cast polycrystals, the GND strengthening is generally achieved at the cost of ductility[16,21,22], since gradient microstructures usually cause inhomogeneous distributions of stress and plastic strain, further enhancing damage localization[23]. In contrast, since the damage could be delocalized by tailoring stress and strain distributions through altering AM-induced gradient microstructures, the AM SXs show potential to overcome the strength-ductility trade-off. Nevertheless, the effects of AM-induced gradient microstructures surrounding columnar dendrites remain largely unexplored in AM alloys, primarily due to the significant interference from orientation distributions in conventional polycrystals, which makes it difficult to quantitatively and explicitly characterize the dendrite-scale gradient microstructures unique to AM[24].

Furthermore, the AM technique offers more flexibility in tailoring mechanical properties[9,25,26]. By tuning the average and amplitude of AM-induced gradient microstructures, it holds promise for achieving various combinations of strength and ductility, making it adaptable to diverse in-service conditions. By using various heat treatment strategies, previous studies on AM polycrystals have achieved improved ductility at the cost of strength through eliminating initial dislocations, while enhancing strength at the expense of ductility via the precipitation of secondary phases during heat treatment[27,28]. However, due to the absence of a quantitative and explicit description of AM-induced gradient microstructures, there are no effective methods to continuously and efficiently tailor mechanical properties of AM alloys.

In this work, we fabricate an SX alloy with typical AM-induced periodic dendrite-scale gradient microstructures, using the electron beam powder bed fusion (EB-PBF) technique. The high preheating temperature enabled by the high energy conversion efficiency, and near-vacuum environment, help maintain a favorable thermal field for directional solidification while effectively suppressing stray grain formation, making EB-PBF more suitable for fabricating SX alloys than other AM methods such as laser powder bed fusion or directed energy deposition. We leverage a unified trigonometric function to describe all gradient microstructures in AM SXs and unveil their effects on mechanical properties, thereby enabling the programmable performance through trigonometric gradient microstructures (TGMs).

## Results and discussion
### Trigonometric gradient microstructures in AM SXs
The EB-PBF technique is employed to fabricate the Ni-based SX alloys (Fig. S1). The as-printed SXs are grain boundary-free and crack-free on a large scale. The primary dendrites are evenly distributed, with an average spacing of 15 μm. The AM SXs exhibit regular and periodic gradient microstructures distributed across dendrite cores to inter-dendrites, including dislocation densities/residual stresses, γ′ precipitate area, γ channel width, and elemental concentrations (Fig. 1a–j), which are referred to as #1–#4 types of TGMs.

As depicted in Fig. 1b and e, a cosine-type trigonometric function is employed to explicitly describe the four types of gradient microstructures within a single period. For a gradient microstructure with type $i$-th, the cosine-type function $\varphi_i(x)$ satisfies the symmetry boundary conditions at the centers of the dendrite core and inter-dendrite, i.e., $\partial\varphi_i(x)/\partial x = 0$. The $A_i$ and $B_i$ in the expression of $\varphi_i(x)$ govern the amplitude and average of TGMs, respectively. In Fig. 1e, we classify two kinds of TGMs with cosine-shaped ($A_i > 0$) and inverted cosine-shaped ($A_i < 0$) distributions. By using the trigonometric function, the averages and amplitudes of TGMs can be quantitatively regulated, facilitating quantifying the role of TGMs, and enabling programmable mechanical properties. As demonstrated in Fig. S2, compared with other formulations such as Gaussian function, the trigonometric function offers advantages in capturing the boundary conditions at dendrite cores, and modeling multiple dendrites through a unified and concise equation, without compromising fitting accuracy.

Induced by thermal stresses and drag force from the liquid during rapid solidification, significant initial dislocations and residual stresses with gradient distributions are present around dendrites in AM SXs (Fig. 1f–j), which is referred to as type #1 of TGMs. The grain reference orientation deviation (GROD) map displays a regular distribution of residual plastic deformation (Fig. 1f), suggesting that the residual plasticity is localized at inter-dendrites. The direct observations of local dislocation densities (Fig. 1i, j) align well with the trend in the GROD map, where the inter-dendrite exhibits a higher dislocation density than dendrite core, resembling the features in typical AM-induced dislocation cellular[29,30]. As a result, the initial dislocation densities exhibit an inverted cosine-shaped distribution. To accommodate the initial density-graded dislocations, the inter-dendrites undergo residual tensile plastic deformation (Fig. S6), while the dendrite cores exhibit minimal residual plastic deformation, as supported by the direct measurement of residual stresses around the dendrites[31]. Under the residual plastic strain with inverted cosine-shaped distribution, to maintain the compatible deformation, there are compressive and tensile residual stresses along building direction (BD) at inter-dendrites and dendrite cores, respectively.

The γ′ precipitate area and γ channel width also display spatial gradient distributions (Fig. 1d), which are indexed as types #2 and #3 of TGMs. The statistical results demonstrate that both γ′ area and γ channel width follow inverted cosine-shaped distribution (Fig. 1g, h), attributed to the difference in solidification rates and local chemical compositions[32]. Moreover, the concentrations of solid-solution elements as type #4 of TGMs, similarly show pronounced gradient distributions (Fig. 1a and c), which are caused by the high cooling rate during the AM process, resulting in a non-equilibrium state[33,34]. The Al/Ti elements are concentrated at inter-dendrites, whereas the Cr/Co elements are localized at dendrite cores, resulting from the solute partitioning at the interface of solid and liquid during the rapid solidification[34,35]. As demonstrated in Fig. S5, the AM-induced TGMs exhibit superior thermal stability after 10-h heat exposure at 700 °C and 900 °C.

### Effects of TGMs on deformation behavior
To experimentally reveal TGMs' effects on deformation behavior, a dual-scale high-temperature in-situ digital image correlation (DIC) technique is developed, in which the millimeter-scale and dendrite-scale strains can be mapped simultaneously. The tensile force is applied parallel to BD, and the applied temperatures include room temperature (RT), 760, 850, 900, and 980 °C. The loading type and applied temperatures are selected based on the anticipated applications of Ni-based SX alloys[5]. The specimens have undergone special treatment for the high-temperature dual-scale DIC analysis (Fig. S7). As depicted in Fig. 2a, the strength decreases significantly after eliminating TGMs using heat treatment at 1050 °C for 10 h (Fig. S17), followed by an increased elongation across all testing temperatures, highlighting the impacts of TGMs on mechanical properties. Meanwhile, from Fig. 2b, when the AM SX with TGMs is highly strained, the initiation and propagation of slip bands with strain localization are captured at specific temperatures, including RT, 850 °C, and 900 °C. After fracture, dense {111} slip bands can be observed in the vicinity of the main crack. (Fig. S14). Under other tested temperatures and in the absence of TGMs, no slip bands are observed, as the strength of the material is lower than the critical stress of γ′ precipitate shearing for slip band initiation[36].

Our subsequent simulation-based analysis reveals that the amplitudes (without altering the average) of types #3 and #4 TGMs can simultaneously improve strength and elongation, under the presence of initial dislocations and residual stresses (type #1) (Fig. 2c). The degree of ductility improvement is proportional to the tensile elongation, according to the stress-strain responses at different temperatures (Fig. S15). The TGMs are further compared with other types of gradient microstructures in as-cast Ni-based polycrystals. The conventional gradient microstructures improve strength at the expense of ductility, which also require post-treatment to introduce them. In contrast, types #3 and #4 of TGMs incorporated in an in-situ manner during the AM process, can improve both strength and elongation in AM SXs under the presence of strong initial density-graded

dislocations and residual stresses. Notably, the TGMs-induced improvement in elongation is more pronounced in the absence of slip bands at higher temperatures, highlighting the more significant benefits from TGMs in high-temperature applications. The comparison in Fig. 2c implies that the strengthening and ductilizing mechanism of TGMs in AM SXs is distinct from that of conventional gradient microstructures in as-cast polycrystals, which will be elaborated in Section "Understand the effects of TGMs on strength and elongation". Additionally, as demonstrated in Fig. S11, the introduction of TGMs does not necessarily cause significant degradation of other mechanical properties, such as low-cycle fatigue performance.

The TGMs-induced strain partitioning is experimentally unveiled based on the dendrite-scale DIC. For the strain $\varepsilon_{xx}$ along loading

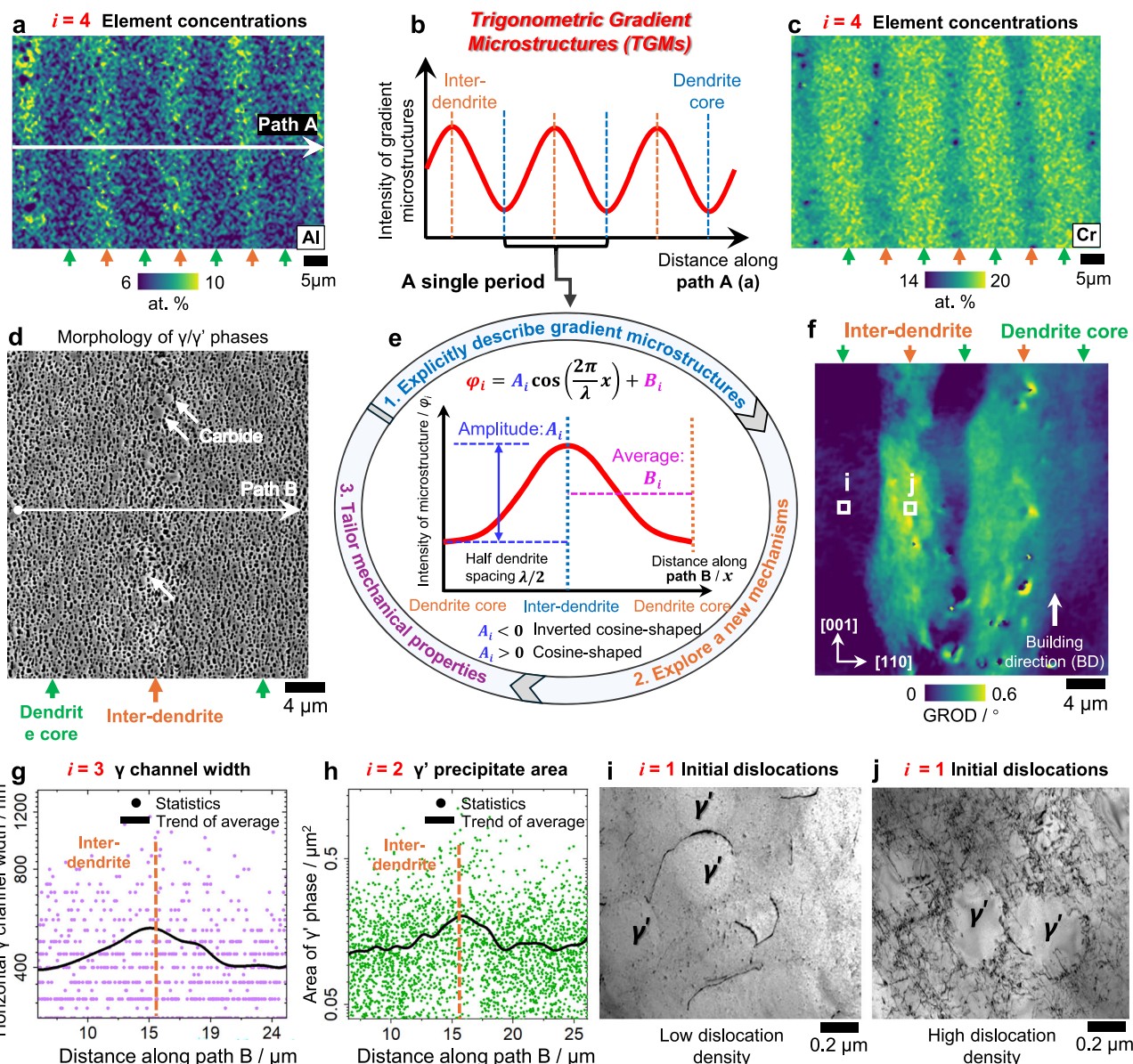

**Fig. 1 | Periodic TGMs in dendrite-scale of AM SX. a, c** Distributions of solid-solution elements Al and Cr (in at. %), which is indexed as type #4 of TGMs. The green and orange arrows indicate inter-dendrites and dendrite cores, respectively. The other solutes with gradient concentration distributions are exhibited in Fig. S4. **b, e** Schematic diagram for the distribution of gradient microstructures across dendrite cores to inter-dendrites. A unified trigonometric expression describing all types of gradient microstructures in AM SXs within a single period, in which the $A_i$ and $B_i$ govern the amplitude and average of $i$-th type of TGM, respectively.

**d** Morphology of γ' precipitates (type #2) and γ channels (type #3) around an inter-dendrite. **g, h** Statistical results for the distributions of γ channel width and γ' precipitate area along path B marked in (**d**). **f** GROD map characterizing the intensity of residual plastic deformation (type #1)[40,55], indicating that the residual deformation is localized at inter-dendrites. **i, j** Transmission electron microscopy (TEM) images for the dislocations at the dendrite core and inter-dendrite, respectively. The γ' precipitates with an L1₂ lattice structure are labeled, while the surrounding regions correspond to the γ (FCC) matrix[56].

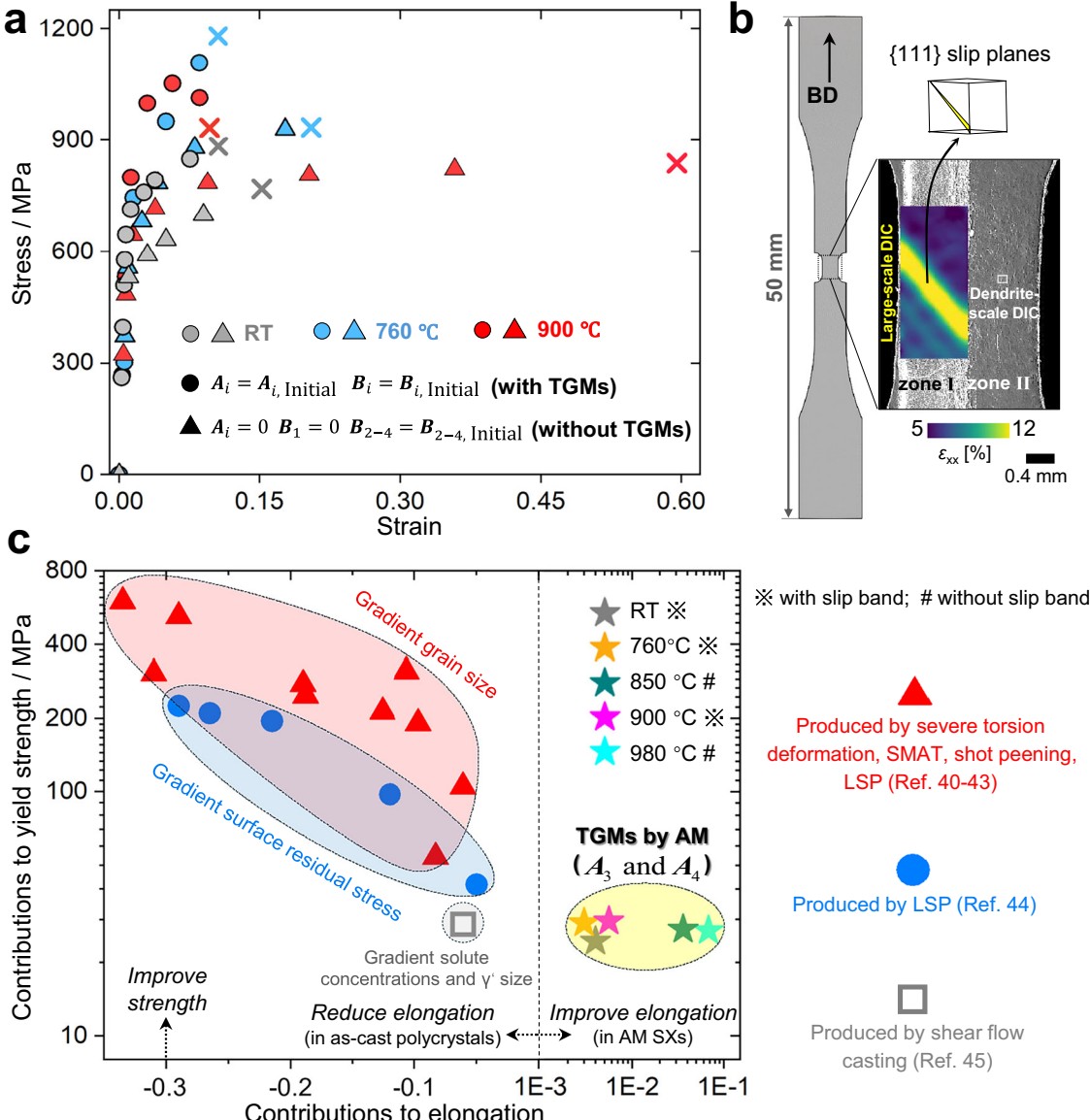

**Fig. 2 | Impacts of TGMs on mechanical properties. a** Comparisons of experimental stress-strain responses in the cases with and without TGMs, in which the definitions of $A_i$ and $B_i$ refer to Fig. 1e. **b** The specimen and dual-scale DIC technique. **c** Comparisons for the contributions to elongation and strength (yield stress corresponding to 2% plastic strain) between beneficial TGMs in AM SXs (amplitudes of type #3 and #4, as identified by simulations), and typical gradient microstructures in as-cast Ni-based polycrystals, including gradient grain size produced by severe torsion deformation[57], surface mechanical attrition treatment (SMAT)[58], shot peening[59], and laser shock peening (LSP)[60]; gradient residual stress produced by LSP[61]; solute segregation/gradient γ channel width produced by shear flow casting[62].

direction, there are notable periodic fluctuations in the strain distributions within slip bands (Fig. 3c, d). To highlight the correlation between TGMs and strain fluctuations, we identify the locations of inter-dendrites through the distribution of carbides (Fig. 3a), since the carbides usually nucleate at inter-dendrites during the AM process[32]. Interestingly, the peaks and troughs of fluctuations regularly occur at the dendrite cores and inter-dendrites, respectively. We further plot the strain distribution along path B-B' within a slip band, indicating that the strain partitioning becomes more pronounced at an increased global strain (Fig. 3k). Nevertheless, at the early stage of plastic deformation in the case without slip bands (Fig. 3b), there is no $\varepsilon_{xx}$ partitioning correlated with the locations of dendrites. Although the TGMs constantly take effect before and after slip band initiation, the strain partitioning only occurs within slip bands. After removing TGMs through heat treatment (Fig. S17), there is no strain partitioning and interaction between slip bands and dendrites in the absence of TGMs,

and the $\varepsilon_{xx}$ tends to be localized around carbides instead (Fig. 3i). Hence, the case without TGMs exhibits a strain pattern distinct from the one with TGMs, highlighting the impact of TGMs on micro-scale deformation behavior.

The distributions of initial dislocations and residual stresses can be inferred from the patterns of lateral strain $\varepsilon_{yy}$. As depicted in Fig. 3e–h, in the case with TGMs and without slip bands, the dendrite cores show a higher absolute value of $\varepsilon_{yy}$, particularly at the early stage of deformation, in which the deviation of $\varepsilon_{yy}$ between the dendrite core and inter-dendrite is nearly 10% of the global strain $\bar{\varepsilon}_{yy}$. After removing TGMs, the $\varepsilon_{yy}$ pattern exhibits no correlation with the dendrite locations (Fig. 3j and l). This observation validates the assumed distributions of initial dislocations and residual plastic strain (Fig. S6). Because, compared with inter-dendrites, the dendrite cores undergo plastic deformation earlier, owing to the tensile residual stress and lower dislocation hardening degree at dendrite cores. Since plasticity

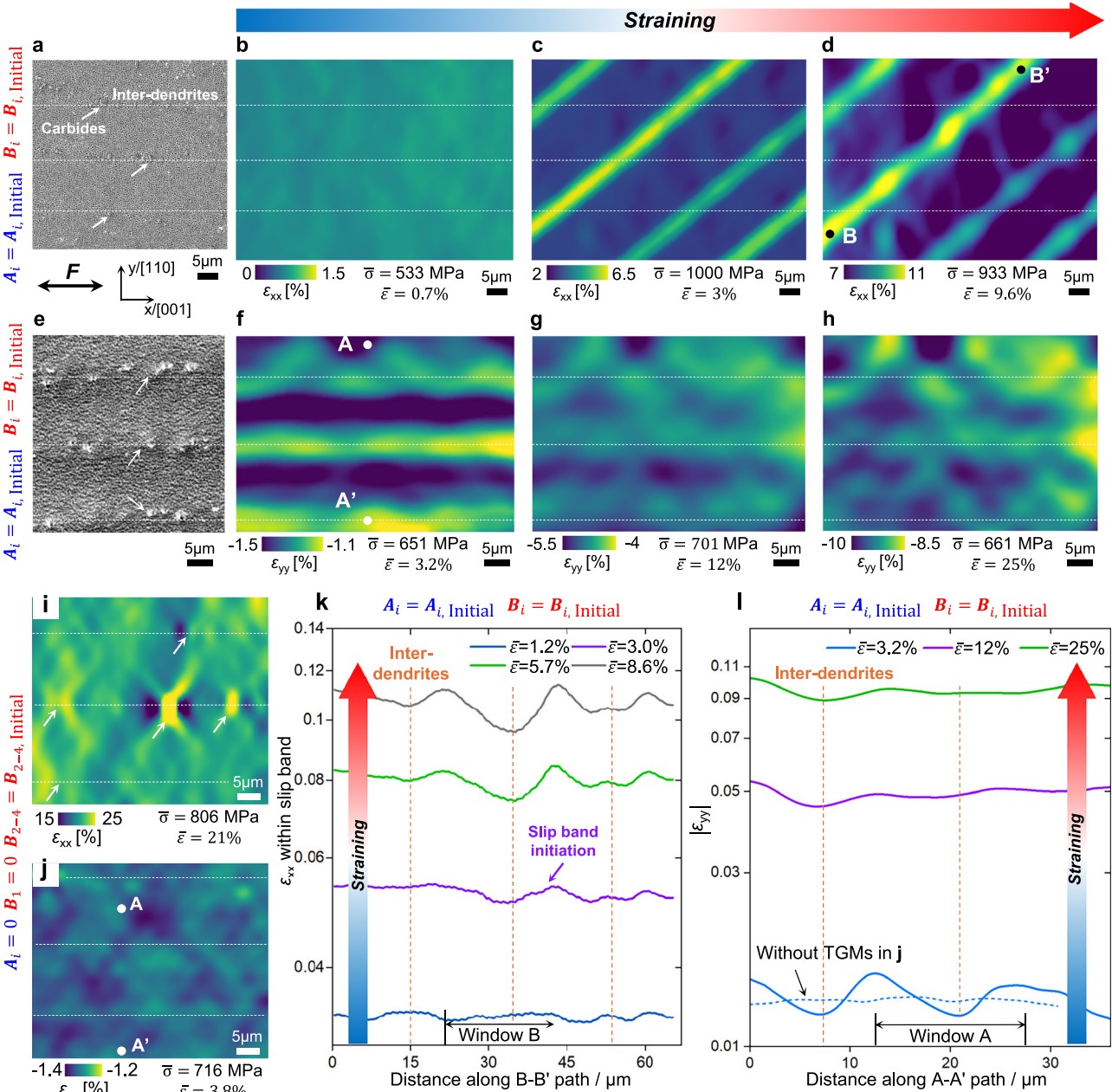

**Fig. 3 | TGMs-induced strain partitioning behavior at dendrite-scale. a–d** TGMs-induced $\varepsilon_{xx}$ distribution and evolution in the case with slip bands at 900 °C, suggesting that the $\varepsilon_{xx}$ within slip bands is localized at dendrite cores. The $\bar{\varepsilon}_{yy}$ denotes global strain. **e–h** TGMs-induced lateral strain $\varepsilon_{yy}$ distribution and evolution in the case without slip bands at 980 °C, indicating that a higher magnitude of $\varepsilon_{yy}$ is concentrated at dendrite cores. **i, j** $\varepsilon_{xx}$ and $\varepsilon_{yy}$ distributions in the case without TGMs at 900 °C, showing $\varepsilon_{xx}$ localized at carbides instead, and indicating an absence of $\varepsilon_{yy}$ localization correlated with dendrite positions. **k** TGMs-induced $\varepsilon_{xx}$

distribution along path B-B' marked in (**d**), suggesting that the strain partitioning degree of $\varepsilon_{xx}$ increases with global strain levels. Logarithmic y-axis is applied to clearly show the strain partitioning at low global strains. **l** TGMs-induced $\varepsilon_{yy}$ distribution along path A-A' marked in (**f**), where the strain partitioning degree of $\varepsilon_{yy}$ decreases with global strain levels. The lateral strain from the case without TGMs in (**j**) is plotted for comparison, which demonstrates that no lateral strain partitioning occurs in the absence of TGMs.

exhibits a stronger Poisson effect than elasticity in SXs, the plasticity-induced $\varepsilon_{yy}$ with a higher absolute value at dendrite cores causes the lateral strain partitioning in Fig. 3f. Moreover, the localization of $\varepsilon_{yy}$ tends to be diminished with the increase of global strains (Fig. 3l). The larger applied deformation diminishes the influence of initial gradient hardening and residual stress on the lateral strain distribution. Because the dendrite cores and inter-dendrites tend to exhibit similar normal plastic strain at the later stage of deformation, resulting in a closer $\varepsilon_{yy}$ due to the comparable Poisson effect at dendrite cores and inter-dendrites.

**Understand the effects of TGMs on strength and elongation**

The crystal plasticity finite element (CPFE) modeling is performed to understand the strain partitioning behaviors for both $\varepsilon_{xx}$ and $\varepsilon_{yy}$ at the dendrite-scale, with a detailed modeling process and validation provided in Section "Crystal plasticity modeling" and Supplementary Methods 1. In the cases without slip bands (Supplementary Movie 2), the simulated $\varepsilon_{xx}$ is uniformly distributed (Fig. S13a) throughout the deformation, while the $\varepsilon_{xx}$ is localized at the dendrite core in the cases with slip band (Fig. 4c and Supplementary Movie 1), aligning well with experimental observations in Fig. 3a–d. Under the TGMs-induced total

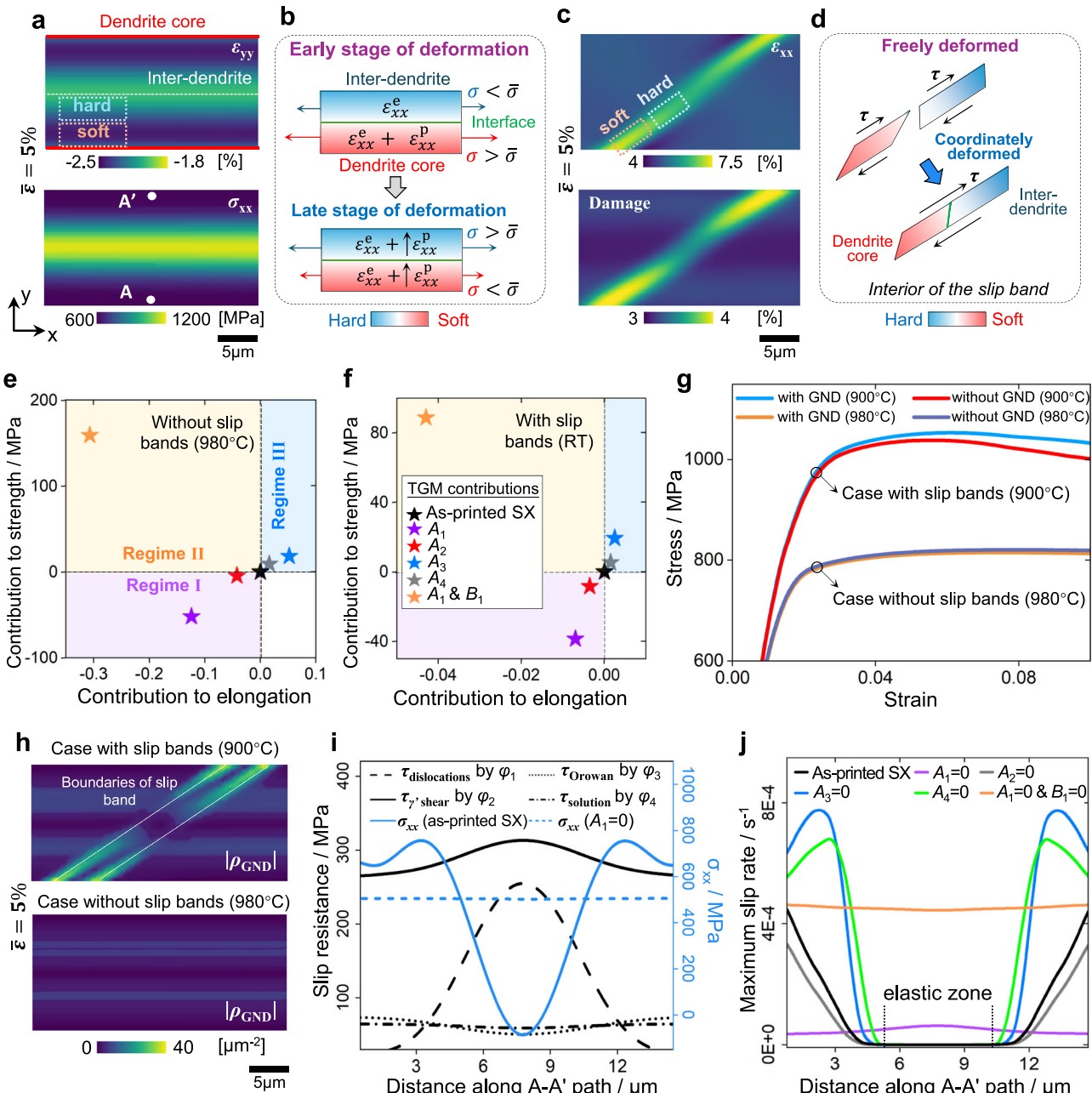

**Fig. 4 | Role of TGMs' influence on mechanical properties. a–d** Simulated strain/stress/damage distributions and the TGMs-mediated strain partitioning mechanisms in the cases with/without slip bands at room temperature and 980 °C, corresponding to loads *m'* and *n* in Fig. S10. **e, f** Contributions of each type of TGMs on strength (corresponding to 2% plastic strain) and elongation, where the simulated stress-strain curves in different cases are shown in Fig. S10. The $A_i$ and $B_i$ represent the amplitudes and averages of the TGM with type *i*, as illustrated in Fig. 1e. These quantitative results are taken from CPFE simulations. **g** Effects of TGMs-induced geometrically necessary dislocations (GNDs) on stress-strain responses in the cases with/without slip bands. **h** Simulated GND density distribution in the cases with/without slip bands. **i, j** Slip resistance, stress, and slip rate distributions along path A-A' in the early stage of deformation (load *m* in Fig. S10a).

slip resistance with inverted cosine-shaped distribution (Fig. 4i), the inter-dendrite is harder than the dendrite core. The interface for deformation coordination takes effects at primary (larger) and secondary (narrower) boundaries in the cases with/without slip band (Fig. 4b and d), respectively. Based on Saint Venant's Principle[37], the deformation coordination has minimal effect in the case with slip band, as the narrower interface between soft and hard regions leads to limited affected zone. Moreover, the damage-induced softening at the dendrite core (Fig. 4c) also contributes to the partitioning of $\varepsilon_{xx}$ within the slip bands. For the lateral strain partitioning (Fig. 4a), under the residual stress with a cosine-shaped distribution, the dendrite core undergoes plasticity first (Fig. 4b), where the additional plastic strain

produces a higher magnitude of $\varepsilon_{yy}$, as the plasticity shows a larger Poisson's ratio than elasticity in SXs. In the later stage of deformation, the $\varepsilon_{xx}$ tends to be uniformly distributed across the dendrite core to inter-dendrite, which causes a more uniformly distributed $\varepsilon_{yy}$. This reproduces the experimental observation in Fig. 3l, where the partitioning degree of $\varepsilon_{yy}$ tends to decrease with the global strains.

In conventional understanding, the strengthening from gradient microstructures depends on the strain gradient-induced GNDs[15,16,20]. Here, in the AM SXs, the contributions of GNDs are quantified through the simulations enabling and disabling GNDs (Fig. 4g). GNDs provide notable extra strengthening only in the cases with slip bands, as the significant strain gradient at the boundaries of the slip band produces

high GND densities. In the absence of slip bands, the GNDs with low densities have a negligible effect on the stress-strain response (Fig. 4h). Compared with the TGMs' effects shown in Fig. 4e, f, the strengthening from GNDs is limited, as the GND-induced improvement in strength is less than 10 MPa. Therefore, the TGMs-induced strengthening cannot be interpreted through the conventional GND-based mechanism.

To understand the TGMs-induced simultaneous enhancement of strength and elongation, we further utilize CPFE simulations to quantify the effects of individual TGMs (Fig. 4e, f). Types #3 and #4 of TGMs in regime III improve strength and elongation simultaneously, whereas types #1 and #2 in regime I have negative effects for both. As shown in Fig. 4i, types #3 and #4 produce slip resistance with a cosine-shaped distribution along path A-A' (defined in Fig. 4a), while the slip resistance induced by types #1 or #2 follows the inverted cosine-shaped distribution. Regarding the TGMs' effects on strength, owing to the compressive residual stresses at the inter-dendrite, the simulated stress exhibits a cosine-shaped distribution at the early stage of deformation (Fig. S12b). Meanwhile, the total slip resistance dominated by initial dislocation hardening exhibits an inverted cosine-shaped distribution (Fig. 4i). Thus, there is an elastic zone within the inter-dendrite, attributed to low stress and high slip resistance (Figs. 4j and S12c). After enhancing the slip resistance with cosine-shaped distribution by types #3 and #4, the total slip resistance at the dendrite core is enhanced, thereby reducing the slip rate at the plastic zone, which delays the yielding and increases the yield strength. Moreover, although the average gradient microstructure intensities are kept constant across comparative cases, the nonlinear relationship between TGMs and corresponding slip resistance leads to changes in average slip resistance, thereby contributing to the strengthening.

Regarding TGMs' effects on elongation, the stress is re-distributed to follow an inverted cosine-shaped distribution at the later stage of deformation in both cases with and without slip band (Figs. 4a and S13b), resulting from the inverted cosine-shaped distribution of total slip resistance. In the case with slip band, the damage characterized by dissipative energy density is localized at the dendrite core (Fig. 4c), resulting from the local low slip resistance and high slip rate (Fig. 4i, j). The enhancement of slip resistance with cosine-shaped distribution induced by types #3 and #4 TGMs, helps increase the total slip resistance at the dendrite core, thereby reducing the slip rate and mitigating the damage localization. In the case without slip band at higher temperatures with larger elongations, although strains are uniformly distributed (Fig. S13a), the energy-based damage is concentrated at inter-dendrite in the late stage of deformation, resulting from stress localization dominated by initial gradient hardening (Fig. S13d). The incorporation of types #3 and #4 TGMs helps mitigate the localization of slip resistance and stress at inter-dendrite, thereby reducing the damage localization. On the contrary, the TGMs corresponding to the slip resistance with inverted cosine-shaped distribution (type #2) produce the opposite influence.

Consequently, we have identified two classes of TGMs with opposite effects on both strength and ductility (Fig. S20). The slip resistance with cosine-shaped distribution, which is out-of-phase with the initial dislocation density distribution, simultaneously enhances strength and ductility. In contrast, the slip resistance with inverted cosine-shaped distribution, which is in-phase with initial density-graded dislocations, has a detrimental effect on both strength and ductility. Hence, in AM SXs with strong initial density-graded dislocations, the TGMs impact the mechanical properties through the features of slip resistance distributions, rather than GND as conventionally understood[21,38]. This finding suggests that enhancing the TGMs with a cosine-shaped distribution helps overcome the strength-ductility trade-off. As demonstrated in Fig. S16, the proposed mechanism shows limited applicability to as-cast SXs due to the absence of initial density-graded dislocations and other gradient

microstructures around dendrites. The TGMs-induced exceptional performance is imparted by the AM process.

## Tailor mechanical properties by tuning TGMs

The role of TGMs in mechanical properties summarized in Fig. S20 is experimentally validated in Fig. 5a–c. Here, only types #1 and #3 of TGMs are focused on, as they have been demonstrated to have the most significant effects on mechanical properties (Fig. 4e, f). To produce TGMs with different intensities, varying scanning speeds are utilized during the AM process for comparison, where the increased scanning speed enhances both initial density-graded dislocations and amplitude of γ channel width distribution (Fig. 5a), owing to the higher thermal gradient and cooling rate. As shown in Fig. 5b, c, the enhanced out-of-phase relationship between initial dislocations and slip resistance of type #3 TGMs, is capable of simultaneously enhancing strength and elongation at a higher scanning speed under both RT and elevated temperature. Additionally, the degree of elongation improvement is larger at the elevated temperature, aligning well with the trend shown in Fig. 2c.

We further develop an explicit approach to efficiently tailor the strength and ductility of AM SXs by tuning TGMs. First, to link the mechanical properties with the intensities of individual TGMs, we conducted high-throughput simulations under various intensities of TGMs. A dimensionless factor $k_i$ characterizing TGMs' intensities is defined in Fig. 5d, in which the definition of $k_1$ for type #1 TGM differs from that of $k_2$–$k_4$. Since we use heat treatment to alter the TGMs, the amplitude for the distribution of initial dislocation cannot be tailored individually, without influencing the average dislocation densities (Fig. 5g). Nevertheless, regarding the γ' precipitate area, γ channel width, and elemental concentrations for types #2 to #4, heat treatment can reduce the amplitude, without influencing the average, as demonstrated in Fig. S17.

As depicted in Fig. 5e and h, based on high-throughput simulation results, both strength $\tilde{\sigma}_i$ and elongation $\tilde{\varepsilon}_i$ change linearly with parameters $k_i$, under the action of $i$-th type of TGMs alone:

$$\begin{cases} \tilde{\sigma}_i = (\tilde{\sigma}_{1,i} - \tilde{\sigma}_{0,i})k_i + \tilde{\sigma}_{0,i} \\ \tilde{\varepsilon}_i = (\tilde{\varepsilon}_{1,i} - \tilde{\varepsilon}_{0,i})k_i + \tilde{\varepsilon}_{0,i} \end{cases} \qquad (1)$$

where $\tilde{\sigma}_{1,i}$ and $\tilde{\sigma}_{0,i}$ denote the strength at $k_i = 1$ and $k_i = 0$ (similar definitions for $\tilde{\varepsilon}_{1,i}$ and $\tilde{\varepsilon}_{0,i}$). Then, we assume that the influence of $k_1$–$k_4$ on strength and elongation follows the principle of linear superposition:

$$\begin{cases} \tilde{\sigma} = \tilde{\sigma}_{initial} - \sum_{i=1}^{4}(\tilde{\sigma}_{1,i} - \tilde{\sigma}_i) \\ \tilde{\varepsilon} = \tilde{\varepsilon}_{initial} - \sum_{i=1}^{4}(\tilde{\varepsilon}_{1,i} - \tilde{\varepsilon}_i) \end{cases} \qquad (2)$$

where $\tilde{\sigma}_{initial}$ and $\tilde{\varepsilon}_{initial}$ are the initial strength and elongation for as-printed SXs. Based on Eqs. (1) and (2), we can map the strength and ductility under arbitrary combinations of TGMs' intensities (Fig. 5f and i).

Furthermore, the magnitude of GROD peaks at inter-dendrites evolves with the heat treatment durations (Fig. 5g), owing to the diffusion and annihilation of initial dislocations at high temperatures[39]. Since the GROD characterizes the plastic deformation degree and related dislocation density[40–42], the intensity of GROD is assumed to follow a proportional relationship with $k_1$. Then, we extract the $k_1$ at different heat treatment durations from the GROD in experiments, by comparing the current GROD with the initial GROD in as-printed SXs. The $k_1$ gradually decreases with heat treatment duration and eventually approaches 0 at 12 h. Meanwhile, due to the gradual microstructure evolution from a non-equilibrium state to an equilibrium state during heat treatment, the $k_3$ characterizing the amplitude of gradient γ channel width is another tunable parameter using heat treatment (Fig. S17). The trajectories displayed in Fig. 5f and i represent

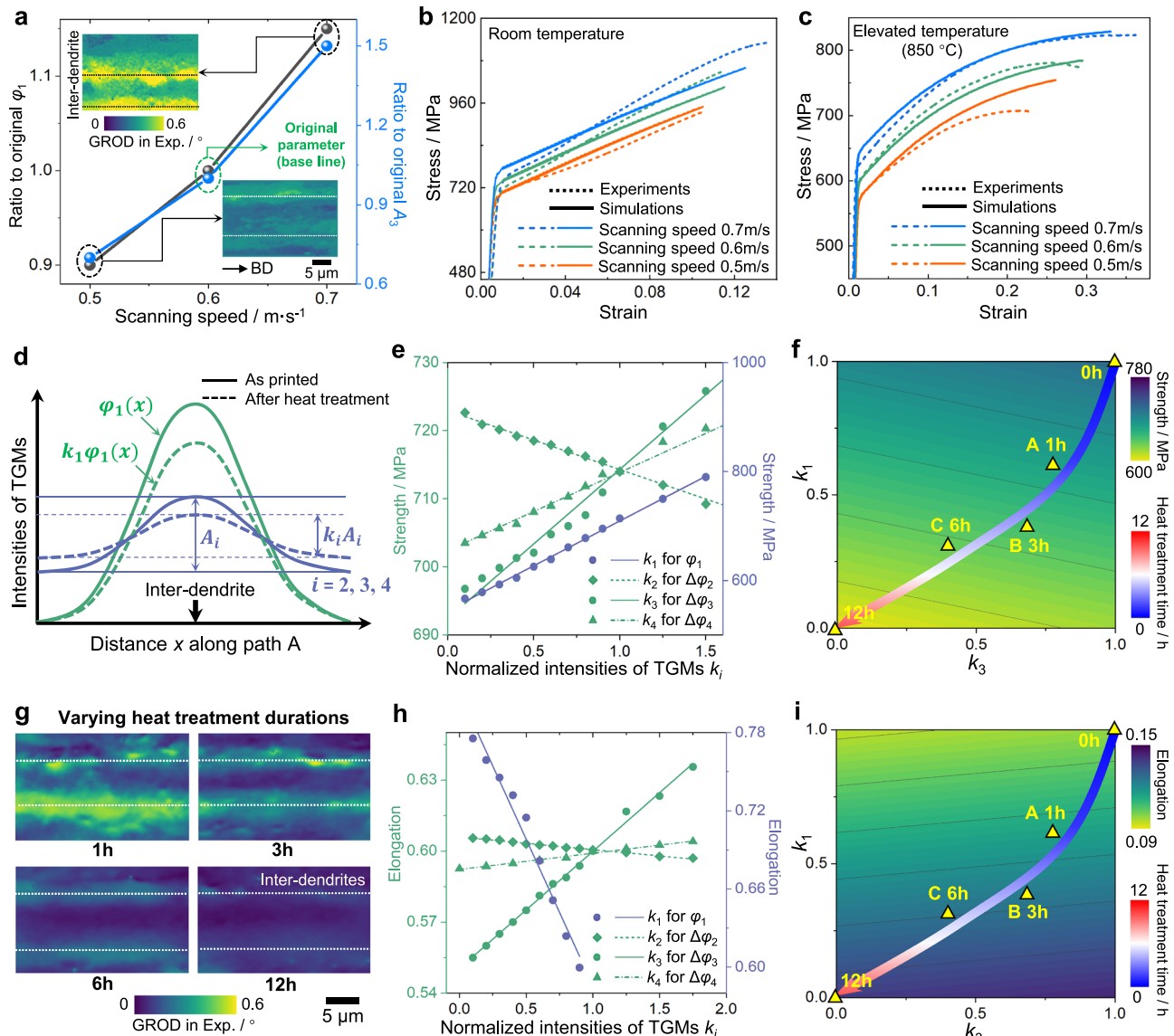

**Fig. 5 | Programmable strength and elongation by tuning TGMs. a** Tuning type #1 and #3 TGMs by varying scanning speeds in the AM process. Compared with the original ones (Fig. 1**f**), higher scanning speed produces TGMs with increased intensities. **b**, **c** Simultaneously improve strength and elongation at both RT and 850 °C, by enhancing the TGMs through increased scanning speeds. **d** Definition of the parameter $k_i$ of the TGM with type $i$-th for tailoring TGMs' intensities using heat treatment at 1050 °C. **e**, **h** Strength and elongation versus $k_i$ at 980 °C, where the scatters are extracted from high-throughput simulation results, accompanied by the linear-fitted curves. **f**, **i** Mapping the strength and elongation against $k_1$ and $k_3$ at RT, along with the trajectory of heat treatment durations for tailoring mechanical properties within a wide range. **g** GROD at varying heat treatment durations, suggesting that both averages and amplitudes of initial dislocation density significantly decrease with heat treatment durations.

the combinations of $k_1$ and $k_3$ at varying heat treatment durations, along which the strength and elongation can be continuously tailored. Furthermore, the strength-elongation mapping is successfully validated against the experimental stress-stain curves for points A/B/C (Fig. S18).

Through the proposed strategies, we achieve the continuous regulation of strength and ductility for AM SXs, by tuning the heat treatment duration to produce various combinations of TGMs' intensities (Fig. 5f and i). Along the trajectories, we can continuously tailor the strength and ductility over a wide range, from 720 MPa/0.1 (heat treated for 0 h) to 600 MPa/0.139 (heat treated for 12 h). An inverse-tailoring approach is further established in Fig. S19, where the mechanical properties are explicitly correlated with heat treatment durations, facilitating the selection of processing parameters to achieve the expected combination of strength and elongation. In future work, more flexible heat treatment strategies can be implemented to realize customized combinations of strength and elongation within the microstructure-property maps established in this study.

In summary, we apply trigonometric function describing the dendrite-scale gradient microstructures in AM SXs to understand the role of TGMs. Differing from the GND strengthening in conventional understanding, the TGMs-induced improvement of strength and elongation depends on the out-of-phase relationship between the TGMs-induced gradient slip resistance and initial density-graded dislocations. Moreover, strength and elongation are explicitly correlated with TGMs' intensities, through the linear relationship revealed by high-throughput simulations. By tuning printing parameters and heat treatment durations, the TGMs are tailored to achieve tunable mechanical properties across a wide range.

The proposed mechanism holds promise to be applied to other AM alloys with initial density-graded dislocations and gradient

microstructures surrounding columnar dendrites, including AM steel[43], polycrystal superalloys[24], Ti[44], and Al[33] alloys. Among them, the high cooling rates and large temperature gradients similarly impart density-graded dislocations, solute segregation[29,45], and gradient in precipitate size[46] around columnar dendrites within grain interiors. Here, the AM SX is employed as a representative model material that eliminates the interference from orientation distribution, thereby revealing the TGM's effects for the typical gradient microstructures around columnar dendrites in AM alloys. In our future work, the applicability of the TGM mechanism to other AM alloys will be further examined.

## Methods

### Additive manufacturing of single-crystal alloys

The Ni-based SX alloy is manufactured by a commercial EB-PBF machine (QBeamLab 200, QuickBeam, China), with detailed printing parameters provided in Tables S1 and 2 and Fig. S1. The key processing parameters are given as follows. The pre-heating temperature of the powder bed is controlled to be 1050 °C, ensuring a stable temperature field for SX growth. A 90° layer-wise rotation is applied to maintain a consistent secondary dendrite growth direction and a cyclically stable temperature gradient. Additionally, to maintain a temperature gradient favorable for SX growth along BD, the baseline scanning speed is selected to be 0.6 m/s through trial-and-error experiments. The selected printing parameters maximize the size of SX within the printed bulks. The heat treatment on as-printed specimens is performed under a split tube furnace (OTF1200X, KJ GROUP, China), with a temperature increasing rate of 5 K/min.

### Microstructure characterization

The morphology of γ/γ′ phases around a dendrite is captured by scanning electron microscope (SEM) (MAGNA, TESCAN, Czech). The orientation distributions are mapped using electron backscatter diffraction (Symmetry2, OXFORD, UK), operating at 20 kV with step sizes of 0.1 μm. The samples used for EBSD analysis are polished using argon ion milling (IM4000II, Hitachi, Japan) at an accelerating voltage of 3–5 keV. The GROD distributions are processed using ATEX software. The dislocation densities at the dendrite core and inter-dendrite are characterized using transmission electron microscopy (JEM-2100F, JEOL, Japan) in the bright-field mode at a voltage of 200 kV, where the samples (4 μm × 4 μm) at inter-dendrites and dendrite cores are prepared using the focused ion beam (Thermo Scientific Scios 2). The statistics method on the γ′ precipitate area and γ channel width are detailed in Fig. S3. An electron probe microanalysis (EPMA) (1720H, Shimadzu, Japan) is utilized to map the concentrations of solid-solution elements under accelerating voltage of 15 kV and step size of 0.1 μm.

### Digital image correlation

The DIC technique is employed to map the strains in both large and dendritic scales, through the correlation analysis of speckle patterns before and after deformation. Within the vacuum chamber of SEM, an in-situ tensile tester (MINI-MT5000, QiYue, China) with a ceramic heater is used to apply deformation and high temperatures, under quasi-equilibrium tension (2 μm/s). The resolution of the SEM images used in DIC analysis is 2560 × 1440 pixels. ZrO$_2$ powder with an average diameter of 2 μm is utilized to prepare the speckle pattern for millimeter-scale DIC, and chemical solution of $5 \times g$ CuSO$_4$ + 25 ml HCl + 5 ml H$_2$SO$_4$ + 20 ml H$_2$O is used to etch the γ/γ′ phases, which serve as deformation carriers at dendritic scale. The DIC analysis is performed on Ncorr platform[47], with strain step sizes of 7 μm in millimeter-scale DIC, and 0.2 μm in dendrite-scale DIC. Additionally, the tensile tests in Figs. 5b and S18 are conducted under a tensile machine (LE5105, LISHI, China), in which a resistance heating furnace is utilized to apply the high temperatures. A video extensometer is used to measure the strain within the gauge length.

### Crystal plasticity modeling

To understand the role of TGMs, we develop a comprehensive dendrite-scale CPFE model, capable of capturing slip band evolution, damage evolution, and multiple strengthening mechanisms in AM SXs (Supplementary Methods 1). The AM-induced residual stresses and initial dislocation hardening are incorporated by introducing the initial plastic deformation gradient with matched initial dislocation densities (Fig. S21).

The slip rate $\dot{\gamma}^s$ along $s$-th slip system is calculated by the damage-coupled Peirce law for dual phases[36,48]:

$$\dot{\gamma}^s = \sum_{i=1}^{2} f_i \dot{\gamma}_{\text{ref}} \left( \frac{|\tau^s|}{(1-D)\tau^s_{c,i}} \right)^{1/n} \text{sign}(\tau^s) \tag{3}$$

where $\tau^s$ is the resolved shear stress, $n$ is the strain rate sensitivity, $\dot{\gamma}_{\text{ref}}$ is the reference slip rate, $D$ is the damage variable for predicting elongation, and $f_1, f_2$ and $\tau^s_{c,1}, \tau^s_{c,2}$ represent the local volume fractions and total slip resistances for γ and γ′ phases, respectively. The influences of TGMs on deformation are reflected in $\tau^s_c$, which involve multiple TGMs-induced strengthening mechanisms. The strain localization within slip band is captured by the reduction in total slip resistance resulting from the shearing of γ′ phases[36], as detailed in Supplementary Methods 1.

The evolution of damage in Eq. (3) is driven by the dissipated energy density[36,49]:

$$dD = \frac{\boldsymbol{\sigma} : d\boldsymbol{\varepsilon}_p}{E_{\text{cr}}} \tag{4}$$

where $\boldsymbol{\sigma}$ and $\boldsymbol{\varepsilon}_p$ are stress and plastic strain tensors, $E_{\text{cr}}$ is the critical dissipation energy. The failure occurs as the $D$ evolves to the threshold $D_{\text{cr}}$, which is validated against experiments in Fig. S15b.

The extra strengthening induced by GNDs is incorporated in the CPFE simulation[50]:

$$\begin{cases} \rho^s_{\text{GNDe}} = -\frac{1}{b} \nabla \gamma^s \cdot \mathbf{n}^s \\ \rho^s_{\text{GNDs}} = \frac{1}{b} \nabla \gamma^s \cdot \mathbf{t}^s \end{cases} \tag{5}$$

where $\rho^s_{\text{GNDe}}$ and $\rho^s_{\text{GNDs}}$ are the edge and screw GNDs, and $\mathbf{t}^s = \mathbf{n}^s \times \mathbf{m}^s$ is the direction of screw dislocation motion[51], where the GNDs contribute to the total dislocation hardening through Eq. (S.5). When implementing Eq. (5), the slip system-dependent accumulated slip $\gamma^s$ is calculated by integrating the slip increments provided by Eq. (3). Subsequently, the edge and screw GND densities are resolved for each slip system based on the gradients of slip in Eq. (5). The presented scalar GND density corresponds to the second norm of the slip system−resolved GND densities.

The representative volume element (RVE) used in CPFE simulation is composed of an inter-dendrite and surrounding dendrite cores (Fig. 6a). Since it has been proven that the dendrite arms (grow normal to [001]) exhibit similar characteristics with dendrite cores in terms of solute concentrations[24,52] and γ/γ′ morphology[53] for SX alloys, the four boundaries parallel to $x$-axis within RVE have the same properties with dendrite cores. The transition region within RVE is designed to avoid interferences from the boundaries where the loads and constraints are applied. The stress-strain responses and distributions are extracted from the simulation results within the gauge region. For the initial conditions of simulations (Fig. 6b), there are compressive residual stresses at the inter-dendrite with positive initial $F^{xx}_{p,0}$, while the tensile residual stresses exist at the dendrite core with zero $F^{xx}_{p,0}$ to maintain the deformation compatibility. The detailed procedures for the pre-

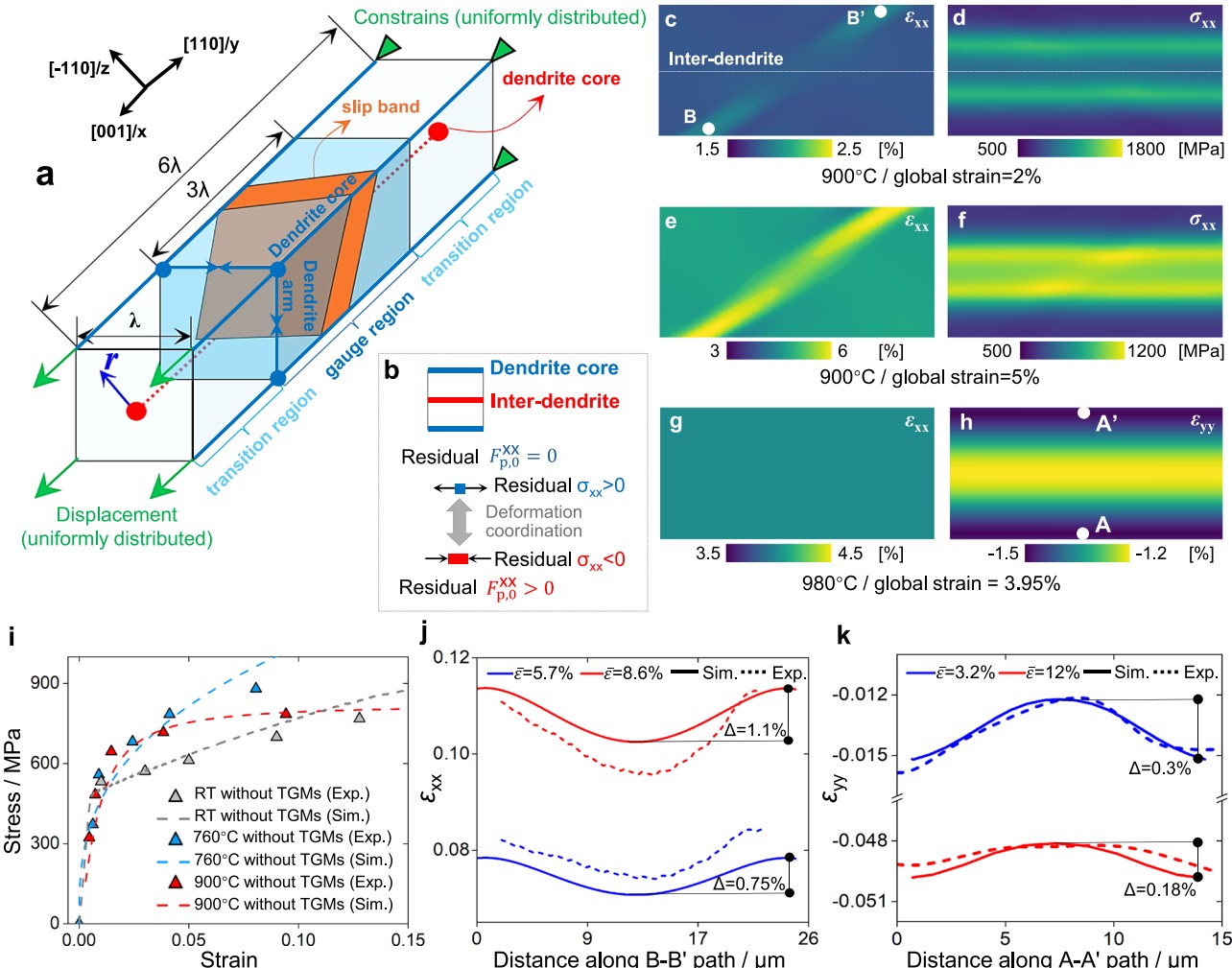

**Fig. 6 | CPFE modeling of dendrite-scale deformation and model validation.**
**a** RVE used in simulations. **b** Directions of residual deformation and residual stresses, where $F_{p,0}^{XX}$ is the dominant $xx$ component within the residual deformation gradient tensor $\mathbf{F}_{p,0}$, which is used to capture the residual stresses, with the detailed demonstration in Fig. S6 and Supplementary Method 1. **c**–**h** Simulated stress/strain distributions within the gauge region of RVE at varying temperatures and global strain levels. **i** Model validation based on the stress-strain curves of SXs without TGMs (heat treated for 12 h). **j** Comparison between experimental and simulated strain $\varepsilon_{xx}$ distribution within slip band, where the experimental strains are extracted from window B in Fig. 3k. **k** Comparison between experimental and simulated lateral strain $\varepsilon_{yy}$ distribution to calibrate the magnitude of residual deformation gradient, in which the experimental strains are extracted from window A in Fig. 3l.

loading of residual deformation and initial dislocations are illustrated in Supplementary Methods 1.

As the input of CPFE simulations, the distributions of TGMs along path $r$ are depicted in Fig. S9. Specifically, the element distributions are determined using EPMA results. The distributions of γ channel width and γ' precipitate area are obtained from the SEM image processing. The assumed initial dislocation density and residual plastic deformation with inverted cosine-shaped distributions, are validated in Fig. S6, with the intensity of $F_{p,0}^{XX}$ calibrated using the DIC results of lateral strain. Moreover, some physics-based parameters are determined from the literature, which are listed in Table S3. The other parameters in the flow rule are calibrated using stress-strain responses of as-printed SX specimens (Fig. S15), and the model is validated against the curves of SX without TGMs (Fig. 6i), in which the experiments and simulations show good agreement.

The key features of dendrite-scale strain distributions revealed by DIC can be reproduced in CPFE simulations. Firstly, the failure modes for the cases with and without slip bands can be well reproduced in simulations (Fig. S14). Besides, when slip bands are absent, although there are gradients of slip resistance, $\varepsilon_{xx}$ is uniformly distributed (Fig. 6g). Differing from the cosine-shaped distribution of residual

stress, the simulated stress after straining is gradually localized at the inter-dendrite (Fig. 6d and f). Stress re-distribution is governed by the gradient of total slip resistance (Fig. 4i), correlated with the negligible partitioning of $\varepsilon_{xx}$ in DIC results. As depicted in Fig. 6c and e, the $\varepsilon_{xx}$ within the slip band is localized at the dendrite core, and the strain partitioning is enhanced in the later stage of deformation (Fig. 6j). The simulated GND density aligns well with the experiment in both pattern and magnitude (Fig. S8). Moreover, the dendrite core shows a higher absolute value of $\varepsilon_{yy}$ (Fig. 6h), with more pronounced partitioning at the early stage of deformation (Fig. 6k). In CPFE simulations, the evolution of both normal and lateral strain distributions aligns well with the DIC results.

## Data availability
All data supporting the findings of this study are available from the corresponding author upon request. Source data are provided with this paper.

## Code availability
The CPFE simulation is performed using an open-source MOOSE framework (https://github.com/idaholab/moose), under the LGPL v2.1

license. The developed code along with a representative simulation case for this study, is available on GitHub (https://github.com/eric199405/CPFE-for-AM-SX), accompanied by a detailed instruction in the README file. A released version used in this paper is deposited in the Zenodo repository[54].

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

## Acknowledgements
This research is supported by the Ministry of Education, Singapore, under its Academic Research Fund Tier 2 (MOE-T2EP50121-0017, W.Y.), the AI Singapore Grant Challenge in AI for Materials Discovery Funding Scheme (AISG2-GC-2023-010, Y.Z. and Y.X.), and the MTC Programmatic funding (M22L2b0111, W.Y., Y.Z., and Y.X.).

## Author contributions
Z.G. conceived the concept, performed simulations, and conducted mechanical experiments. Y.L. and F.L. fabricated the AM SXs. L.F. and S.W. performed microstructure characterization. D.H. and G.P. contributed to data processing. Y.X. processed and analyzed the CPFE simulation results. Y.Z. and W.Y. conceived the concept and revised the manuscript. All authors contributed to the discussion of the data and edited the manuscript.

## Competing interests
The authors declare no competing interests.
