## [Transparent Peer review file · Nature Communications]

Trigonometric Gradient Microstructures in Additively Manufactured Single Crystals Enable Strength-Ductility Synergy and Programmable Performance

Corresponding Author: Dr Yilun Xu

Version 0:

Reviewer comments:

Reviewer #1

(Remarks to the Author)

This study employs a cosine-type function to describe the gradient microstructures in electron beam powder bed fusion (EB-PBF) single-crystal (SX) alloys and establishes a linear relationship between the function parameters (TGMs intensities) and mechanical properties. This approach provides an explanation for the simultaneous enhancement of both strength and ductility. The paper represents an advancement in understanding and utilizing gradient microstructures for advanced materials design. However, several concerns need to be addressed regarding result clarifications, the realization of programmability, and the generalisation of the proposed approach.

1. Could the authors provide a justification for using a cosine-type function to describe gradient microstructures rather than other forms, such as a Gaussian-based distribution? The latter seems more flexible and aligns more closely with the concept of 'programmability.' Is there any physical rationale behind it or just a coincidence? What if we changed to another type of SX? Can we draw the same conclusions. I think it is very important to make this point clear as that's the "CORE" of the manuscript.
2. It is understood that fabricating single-crystal (SX) structures via additive manufacturing is challenging. However, the authors have successfully identified the parameter window for printing. Could the authors provide a detailed explanation of the methodology used for parameter selection?
3. On page 4, the statement "the dendrite core exhibits a higher dislocation density" contradicts the markings in Figure 1-b2 and b3. Please clarify this inconsistency.
4. On page 4, the term "convex distribution" is used. However, "convex" and "concave" typically describe functions rather than distributions. Please clarify.
5. In Figure 1-d1, the caption describes the atomic concentration distribution of Cr and Al, but the numeric order in the figure suggests the opposite. Additionally, Figure S2 refers to Co and Al for elemental segregation. Please note that "atomic" and "elemental" are distinct concepts. Typos also appear in Figure 5e, where "PGMs" is used instead of the correct term. Please carefully proofread the entire manuscript.
6. In Figure 1-c3, the plot represents γ' area data points, whereas the text describes γ' precipitate size. These should be consistent, or the relevance should be clarified. Additionally, the dendrite separation marks in c1 (tilted) differ from those in other subfigures (straight), despite using similar length scales. What is the basis for calculating the statistical data presented in c2 and c3?
7. On page 6 (Section 2.2), the statement "tensile force is applied parallel to the BD and dendrite growth" may not be entirely accurate, as dendrite growth directions vary due to inter-dendrite formations, as acknowledged by the authors. Please refine this statement.
8. The authors demonstrate that (a) AM-processed SX alloys exhibit TGMs and (b) varying processing parameters (e.g., scan speed and heat treatment duration) alters mechanical properties. These facts are true. The establishment of a cosine-type function to correlate TGMs intensity with mechanical properties is novel; however, it raises the question of whether this is more of a finding or an explanation rather than true "programmability." The study primarily discloses and explains that TGMs are responsible for the synchronized enhancement of strength and elongation. To achieve true "programmability," an inverse-tailoring approach may be necessary—where processing parameters are selected based on desired mechanical properties.

9. The authors state, "Our work offers a new pathway to simultaneously enhance strength and ductility through AM..." However, the study focuses on a specific Ni-based SX alloy. A major concern remains: can the TGM tailoring approach be generalized to other material systems?
10. The reviewer has limited knowledge on CPFEA modelling, and thus could not provide a constructive advice on this part.

(Remarks on code availability)

Reviewer #2

(Remarks to the Author)

The submission focuses on generation of trigonometric gradient microstructures in Ni based single crystals fabricated via AM. High throughput simulations and experimental validations were conducted to analyze the mechanical properties of the TGMs and their strength-ductility tradeoff as opposed to what is commonly observed in conventionally heat treated alloys and metallic materials. With the given findings, it is proved that a simultaneous enhancement of strength and ductility can be achieved by taking advantage of the phase relationship between initial density-graded dislocations and TGMs, which helps balancing the stress/strain distribution and lower chances of damage localization within the material (also in agreement with the designed concave slip resistance distribution in the material). The work is interesting and written well. Given the expertise of the reviewer, the major focus in the given comments and evaluation was on mechanical properties. The authors are asked to clarify the points mentioned in the comments below:

- by "post fabrication" do the authors refer to "post treatment" methods of change of material microstructure?
- The reported properties and trends seem interesting and promising. However, can the authors comment on the long-term stability of this? Any potential aging in these architected materials? Especially potential degradation in case of prolonged exposure to elevated temperatures. Please comment on this in the manuscript.
- How can the extent of this improvement in ductility and strength impact other critical properties of the alloys? Especially in case of cyclic loading where any change in material, especially slip system can dramatically impact the life time of the component, can what is seen here as improvement show cause even a higher improvement in fatigue performance? Or the opposite, can introduction of the TGMs negatively impact the cyclic plasticity and formation of persistent slip bands? An elaboration on this in the manuscript would be beneficial.
- Majority of the scientific discussions are relying on the two scale DIC analyses. The authors are asked to provide fractography of the tested specimens, at least on selected cases, in the text. Could you observe any traces of the 111 slip planes on the fracture surface? How did this vary by an increase in the temperature?

(Remarks on code availability)

Reviewer #3

(Remarks to the Author)

This manuscript presents a very interesting and comprehensive study on additively manufactured (AM) single crystals, highlighting the significance of multiple trigonometric gradient microstructures (TGMs) surrounding dendrites. Unlike the conventional geometrically necessary dislocations (GNDs) mechanism, the AM-induced TGMs overcome the strength-ductility trade-off through a unique 'phase relationship'. The authors further demonstrate how to effectively tailor mechanical properties by tuning TGMs. This work is both impressive and highly promising, with the potential to attract widespread attention in Nature Communications. However, in prior to the decision, the following comments should be addressed carefully.

Major comments:

1. The authors should provide more details on the statistics of γ channel width and γ' phase size in Fig. 1c, as these are critical components of TGMs.
2. Since GND-induced extra strengthening is referred to as a conventional mechanism, it would be beneficial to calculate the GND density from DIC analysis in Fig. 3a–b. This data could then be used to validate the simulated GND density presented in Fig. 4d.
3. In Fig. 2b, the data on conventional gradient microstructures in as-cast polycrystals is limited. The authors should provide additional data from the literature to better support the advancements of TGMs.
4. In Fig. 3e–f, it would be helpful to plot the strain distributions for the samples without TGMs along the B–B' and A–A' paths. This would more explicitly highlight the effects of TGMs on the partitioning of normal strain and lateral strain.
5. Regarding the validation of the proposed CPFEE model in Fig. 6g, only a single global strain level is used to validate the simulation results for lateral strains. It is recommended to incorporate additional global strain levels to further validate or calibrate the model.

Minor comments:

1. Eq. (5) is the definition formula of GND. The authors should further clarify how to implement GND in CPFEE simulations.
2. In Fig 6 a, please provide the definition of $F_{p,0xx}$ in the caption.

(Remarks on code availability)

Version 1:

Reviewer comments:

Reviewer #1

(Remarks to the Author)

The authors have properly addressed all the questions raised by the reviewer. It can be accepted in the current form.

(Remarks on code availability)

Reviewer #2

(Remarks to the Author)

The authors have successfully implemented the required and suggested modifications in their revised manuscript. I would especially applaud the authors for going beyond and performing additional experiments to support their findings in response to the given comments (specifically, thermal stability tests, low-cycle fatigue tests, and fractography). Given this, I recommend publication of this submission in its current form.

(Remarks on code availability)

Reviewer #3

(Remarks to the Author)

The author team has answered my concerns, and I agree to accept this paper.

(Remarks on code availability)

None

Submission NCOMMS-25-10755

Response to Reviewers' Comments

**Trigonometric Gradient Microstructures in Additively Manufactured Single Crystals
Enable Programmable Exceptional Performance**

Dear Editor and Reviewers,

We sincerely thank the reviewers and editor for their detailed feedback on our submitted manuscript. We have made substantial revisions according to the reviewers' and editor's comments. All points raised by the reviewers/editors have been addressed, with our responses detailed below. The modifications to the manuscript addressing the reviewers' comments have been highlighted in red in the revised manuscript. We hope that the revised version of the manuscript will now be considered suitable for publication in ***Nature Communications***.

Regards,

Dr. Yilun Xu
Senior Scientist
Institute of High Performance Computing
A*STAR, Singapore

Prof. Wentao Yan
Associate Professor
Department of Mechanical Engineering
National University of Singapore, Singapore

Reviewer #1:

General comments: This study employs a cosine-type function to describe the gradient microstructures in electron beam powder bed fusion (EB-PBF) single-crystal (SX) alloys and establishes a linear relationship between the function parameters (TGMs intensities) and mechanical properties. This approach provides an explanation for the simultaneous enhancement of both strength and ductility. The paper represents an advancement in understanding and utilizing gradient microstructures for advanced materials design. However, several concerns need to be addressed regarding result clarifications, the realization of programmability, and the generalisation of the proposed approach.

Response: We sincerely thank the reviewer for the positive comments. We have incorporated the suggested modifications in the revised version, especially for result clarifications, realization of programmability, and generalisation of the proposed approach.

Comment 1: Could the authors provide a justification for using a cosine-type function to describe gradient microstructures rather than other forms, such as a Gaussian-based distribution? The latter seems more flexible and aligns more closely with the concept of 'programmability.' Is there any physical rationale behind it or just a coincidence? What if we changed to another type of SX? Can we draw the same conclusions. I think it is very important to make this point clear as that's the "CORE" of the manuscript.

Response: Thanks for the insightful comment. In the revised manuscript and Supplementary Materials, we have systematically compared the fitting performance of the trigonometric function and Gaussian function for the intensities of TGMs. The fitting results indicate that both functions align well with statistical results in experiments, similarly exhibiting high correlation coefficients. However, the non-periodic nature of Gaussian function limits it to capture the periodic distribution spanning multiple dendrite structures, which is the key feature of gradient microstructures in AM SXs, whereas this feature can be effectively represented using a unified and concise trigonometric function. Moreover, as the derivative of a Gaussian function is nonzero everywhere, the Gaussian function fails to describe the zero-slope characteristic observed at the dendrite cores, which arises from the inherent symmetry of gradient microstructures with respect to dendrite cores formed during the solidification process¹. Consequently, the trigonometric function is adopted in the present work. A detailed explanation is provided in **Fig. S2** within Supplementary Materials:

Fig. S2 Comparison between trigonometric fitting and Gaussian fitting: (a) Al distribution (in at. %) characterized using EPMA, which is employed as an example to test different fitting functions. (b) Distribution of Al atomic percentage along path A marked in a, in which each scatter is determined as the average within the height of EPMA domain. (c-d) Fitting the experimental Al distribution within window A using trigonometric and Gaussian functions, respectively. (e) Distribution of γ matrix channel width surrounding an inter-dendrite

(statistics approach detailed in **Fig. S3**), fitted using trigonometric and Gaussian functions. The correlation coefficients R^2 indicate that both functions exhibit comparable accuracy, similarly showing good agreement with the statistical results in element distribution and γ/γ' morphology. However, since the derivative of a Gaussian function is nonzero everywhere, the Gaussian function fails to capture the boundary condition of $dy/dx \approx 0$ at dendrite cores, where the plateaus of gradient microstructure intensities with $dy/dx \approx 0$ (marked as the ranges of red double arrows in **b**) consistently exist in experiments. This arises from the inherent symmetry of gradient microstructures with respect to dendrite cores formed during the solidification process¹. In addition, the trigonometric function is capable of representing the gradient microstructures distributed along multiple dendrites through a unified and concise equation, while the non-periodic Gaussian or polynomial functions fail to achieve this. Hence, the trigonometric function facilitates future investigations into the variations in the dendrite spacing and intensities of gradient microstructures among multiple dendrites. **(f)** An application case of damped trigonometric function² on multiple dendrites with varying peak height. Notably, since the element segregation shows a strong correlation with initial dislocations^{3,4} and formation of γ' phases^{5,6} governed by local chemical compositions, the above trends could be similarly applicable to other types of gradient microstructures around dendrites, since the plateaus with $dy/dx \approx 0$ also can be observed at dendrite cores for the precipitate area in **Fig 1 c3**.

In the 2nd paragraph of **Section 2.1** in manuscript, we have added that: **As demonstrated in Fig. S2**, compared with other formulations such as the Gaussian function, the trigonometric function offers advantages in capturing boundary conditions at dendrite cores, and modeling multiple dendrites through a unified and concise equation, without compromising fitting accuracy.

Moreover, to illustrate the applicability of TGMs on other types of SX, we have provided the microstructure characteristics of a conventional as-cast SX in **Fig. S16** within Supplementary Materials. Accordingly, in the last paragraph of **Section 2.3**, we have mentioned that: **As demonstrated in Fig. S16**, the proposed mechanism shows limited applicability to as-cast SXs due to the absence of initial density-graded dislocations and other gradient microstructures. The TGMs-induced exceptional performance is imparted by the AM process.

Fig. S16 Benchmark against an as-cast Ni-based SX alloy: (a) Morphology of dendrites under optical microscopy for an as-cast SX (DD6, China), indicating that the columnar dendrite spacing of as-cast SX is nearly 250 μm, more than 15 times larger than that of AM SXs. (b) EBSD-based GROD map for as-cast SX exhibits negligible residual deformation distribution within each dendrite structure. (c) TEM observations on the inter-dendrite of the as-cast SX alloy, suggesting an absence of initial dislocations and associated density gradients. (d) EPMA results for solid-solution elements, showing minimal solute segregation across dendrite cores to inter-dendrites. Therefore, compared with AM SX alloys, as-cast SX alloys exhibit negligible initial dislocation densities and minimal solute segregation. As demonstrated in **Section 2.3**, since the TGM-induced simultaneous enhancement of strength and ductility primarily relies on the presence of initial density-graded dislocations, the proposed mechanism unique to AM SXs could not be applicable to as-cast SX alloys. Given that TGMs are introduced by the high cooling rates and large temperature gradients inherent to the AM process, the TGM mechanism is unique to AM alloys.

References in this comment:

[1] Yu, Y., Li, Y., Lin, F. & Yan, W. A multi-grid Cellular Automaton model for simulating dendrite growth and its application in additive manufacturing. Additive Manufacturing 47, 102284 (2021).

- [2] Fort, A., Landi, E., Moretti, R., Carbone, P. & Moschitta, A. Damped Sine Wave Parameter Extraction: Application to QCM-D Signals for Accurate Measurements. *IEEE Transactions on Instrumentation and Measurement* (2025).
- [3] Guo, B. et al. Segregation-dislocation self-organized structures ductilize a work-hardened medium entropy alloy. *Nature Communications* 16, 1475 (2025).
- [4] Wang, Y. M. et al. Additively manufactured hierarchical stainless steels with high strength and ductility. *Nature Materials* 17, 63-71 (2018).
- [5] Huang, M. et al. Effect of γ forming element additions on the homogenization behavior and formation of hierarchical microstructures in Ni-based superalloys. *Journal of Alloys and Compounds* 975, 172929 (2024).
- [6] Neumeier, S. et al. Advanced Polycrystalline γ' -Strengthened CoNiCr-Based Superalloys. *Metallurgical and Materials Transactions A* 55, 1319-1337 (2024).

Comment 2: It is understood that fabricating single-crystal (SX) structures via additive manufacturing is challenging. However, the authors have successfully identified the parameter window for printing. Could the authors provide a detailed explanation of the methodology used for parameter selection?

Response: Thanks for the valuable comment. The fabrication of EB-PBF SXs relies on the careful selection of key printing parameters, including preheating temperature, scan direction, and scan speed. Firstly, the preheating of the substrate is essential to achieve a powder bed temperature of 1050 °C, ensuring a stable temperature field for SX growth. Furthermore, layer-wise rotation of 90° is applied to maintain consistent growth direction of secondary dendrites. Additionally, the scan speed as a key processing parameter, is carefully selected through experiments to ensure a temperature gradient favorable for primary dendrite growth. As shown in **Fig. R1**, when the scan speed is selected as 0.6 m/s, the SX with the largest size can be obtained within the central areas of the printed bulks. Thus, the experimentally selected preheating temperature, scan direction, and scan speed are adopted as the optimal key printing parameters for the fabrication of the SXs in this study.

We have incorporated **Fig. R1** into **Fig. S1** in Supplementary Materials accordingly, and added the caption that **(e) Selection of printing parameter, indicating that the SX with maximum size can be obtained at the scan speed around 0.6 m/s.**

Fig. R1 The key parameter for the single-crystal formation during printing — scan speed v : (a) $v=3$ m/s, (b) $v=2$ m/s, (c) $v=1$ m/s, (d) $v=0.6$ m/s, (e) $v=0.3$ m/s. The tensile specimens are sampled from the center of printed bulks.

In the 1st paragraph of **Section 3.1**, we have added that: ...with detailed parameters provided in **Table S1-2** and **Fig. S1**. The key processing parameters are given as follows. The pre-heating temperature of powder bed is controlled to be 1050 °C, ensuring a stable temperature field for SX growth. The layer-wise rotation of 90° is applied to maintain a consistent growth direction of secondary dendrites. Additionally, to maintain a temperature gradient favorable for SX growth along BD, the scan speed is selected to be 0.6 m/s through trial-and-error experiments. The selected printing parameters maximize the size of SX within the printed bulks.

Comment 3: On page 4, the statement "the dendrite core exhibits a higher dislocation density" contradicts the markings in Figure 1-b2 and b3. Please clarify this inconsistency.

Response: Thanks for the careful review. The **b2** and **b3** in **Fig. 1** are mislabelled in the original manuscript. The TEM images with high and low dislocation densities should correspond to the regions at the inter-dendrite and dendrite core, respectively. This trend is also consistent with the experimental observations on AM-induced dislocation cellular from literature¹, where the initial dislocations are similarly localized at cellular boundaries

associated with the inter-dendrite region. The **Fig. 1 b** in the manuscript has been corrected, accordingly.

Additionally, we have carefully checked and corrected the labelling of dendrite cores and inter-dendritic regions throughout the manuscript. In this work, the carbides (TiC, NbC) are used to identify the locations of inter-dendritic regions under SEM, as they are prone to nucleate at inter-dendrites², which are the last to solidify and rich in Ti and Nb³. Furthermore, based on the identified positions of dendrite cores and inter-dendritic regions, the distributions of solid-solution elements are also consistent with previous reports on Ni-based alloys^{3,4}. Therefore, in the revised manuscript, the labelling for the locations of dendrite cores and inter-dendrites is convincing and reliable.

Fig. R2 Revised Fig. 1 b in manuscript.

References in this comment:

- [1] Yu, P. et al. Microstructure evolution and composition redistribution of FeCoNiCrMn high entropy alloy under extreme plastic deformation. *Materials Research Letters* 10, 124-132 (2022).
- [2] Wang, R. et al. Microstructure characteristics of a René N5 Ni-based single-crystal superalloy prepared by laser-directed energy deposition. *Additive Manufacturing* 61, 103363 (2023).
- [3] Florian Pixner. et al., Thermal cycling effects on the local microstructure and mechanical properties in wire-based directed energy deposition of nickel-based superalloy, *Additive Manufacturing*, 83, 104066 (2024).
- [4] Ren, N. et al. Solute trapping and non-equilibrium microstructure during rapid solidification of additive manufacturing. *Nature Communications* 14, 7990 (2023).

Comment 4: On page 4, the term “convex distribution” is used. However, "convex" and "concave" typically describe functions rather than distributions. Please clarify.

Response: Thanks for the thoughtful comment. We agree that the terms ‘convex’ and ‘concave’ are more appropriately used to describe the curvature of functions rather than distributions. To avoid confusion, we have changed the wording from ‘concave’ and ‘convex’ into ‘cosine-shaped’ and ‘inverted cosine-shaped’ in both the figure and the main text, with a schematic diagram shown in **Fig. R3**. We believe that this revision more accurately reflects the mathematical context and addresses the reviewer’s concern.

Fig. R3 Schematic diagram for cosine-shaped / inverted cosine-shaped distributions.

Comment 5: In Figure 1-d1, the caption describes the atomic concentration distribution of Cr and Al, but the numeric order in the figure suggests the opposite. Additionally, Figure S2 refers to Co and Al for elemental segregation. Please note that "atomic" and "elemental" are distinct concepts. Typos also appear in Figure 5e, where "PGMs" is used instead of the correct term. Please carefully proofread the entire manuscript.

Response: Thanks for the careful review. The caption of **Fig. 1 d** has been corrected as '(d1-d2) Distributions of solid-solution elements Al and Cr (in at. %), respectively.'. The caption of **Fig. S2 (Fig. S4** in revised Supplementary Materials) has been modified as '**Fig. S4 Distributions of solid-solution elements Co and Ti (in at. %)**'. Meanwhile, the 'PGMs' in **Fig. 5 e** has been corrected as '**TGMs**'.

In addition, we have thoroughly checked and refined the entire manuscript, including the Supplementary Materials to ensure the correctness.

Fig. R4 Revised Fig. 5 e in manuscript.

Comment 6: In Figure 1-c3, the plot represents γ' area data points, whereas the text describes γ' precipitate size. These should be consistent, or the relevance should be clarified. Additionally, the dendrite separation marks in c1 (tilted) differ from those in other subfigures (straight), despite using similar length scales. What is the basis for calculating the statistical data presented in c2 and c3?

Response: Thanks for the constructive comment. To ensure consistency, we have changed all 'precipitate size' into '**precipitate area**' in both manuscript and Supplementary Materials. Additionally, we have changed the marks of the dendrite core and inter-dendrite in **Fig. 1 c1**, without using the tilted curves (**Fig. R2** in this response letter), to ensure consistency with other sub-figures. Moreover, details of the statistical approaches used in **Fig. 1 c2** and **c3** have been provided in **Fig. S3** within Supplementary Materials:

Fig. S3 Statistical methods for precipitate area, width, length, and matrix channel width: (a) Statistics on precipitate dimension and area using ImageJ-Pro Plus software⁷, in which the central coordinates, height, length, and area for each precipitate can be accurately identified and extracted for further analysis. (b) Statistics on the horizontal width of γ matrix channel based on the chord length method for the dual phases⁸, which is implemented based on our previous work⁹. The short blue lines represent the identified horizontal width of the matrix channel. (c) A statistical approach to capture the trend of variation in precipitate dimension and area along path B. For each γ' identified, the statistical results on precipitate area and coordinate are plotted in this figure as scatters. Then, path B is divided into 30 blocks, and statistical averaging is performed within each block. The resulting block-wise averages are then connected to form the 'trend of average' curve.

Besides, in the **3.2 Microstructure characterization**, we have added that: The statistics method on the γ' precipitate area and γ channel width are detailed in **Fig. S3**.

References in this comment:

- [7] ImageJ-Pro Plus (Trial version), <https://imagej.net/ij/index.html> (2025).
 [8] Caccuri, V., Desmorat, R. & Cormier, J. Tensorial nature of γ' -rafting evolution in nickel-based single crystal superalloys. *Acta Materialia* 158, 138-154 (2018).

[9] Guo, Z., Song, Z., Huang, D. & Yan, X. Matrix Channel Width Evolution of Single Crystal Superalloy Under Creep and Thermal Mechanical Fatigue: Experimental and Modeling Investigations. *Metals and Materials International* 28, 2972-2986 (2022).

Comment 7: On page 6 (Section 2.2), the statement "tensile force is applied parallel to the BD and dendrite growth" may not be entirely accurate, as dendrite growth directions vary due to inter-dendrite formations, as acknowledged by the authors. Please refine this statement.

Response: Thank you for the thoughtful review. In this study, although the build direction (BD) and the primary dendrite growth direction are roughly aligned, they are two distinct concepts and exhibit slight differences and deviations. To maintain rigor and clarity, we define the tensile direction as parallel to the BD in this paper. We have revised the relevant statements in **Section 2.2** as: **The tensile force is applied parallel to the BD.**

Comment 8: The authors demonstrate that (a) AM-processed SX alloys exhibit TGMs and (b) varying processing parameters (e.g., scan speed and heat treatment duration) alters mechanical properties. These facts are true. The establishment of a cosine-type function to correlate TGMs intensity with mechanical properties is novel; however, it raises the question of whether this is more of a finding or an explanation rather than true "programmability." The study primarily discloses and explains that TGMs are responsible for the synchronized enhancement of strength and elongation. To achieve true "programmability," an inverse-tailoring approach may be necessary—where processing parameters are selected based on desired mechanical properties.

Response: Thanks for the valuable comment. In this work, we have revealed the role of TGMs in simultaneously enhancing strength and ductility, and demonstrated their ability for tailoring mechanical properties. In the revised manuscript, we have highlighted the novelty mentioned by the reviewer in the conclusion part. Moreover, the microstructure-property relationship has been extended to establish an inverse-tailoring approach, facilitating the selection of heat treatment duration for the expected combination of strength and elongation. Besides, please note that the 'programmable' also means tunable mechanical properties through tailoring the processing parameters, and this definition has been recognized in the journal¹⁰.

In addition, the comparison between simulations and experiments under varying scanning speeds at an elevated temperature of 850 °C has been added in **Fig. 5 b2**. This complement not only reinforces the dependence of mechanical properties on TGMs' intensities and

processing parameters, but also further validates the TGM-induced simultaneous enhancement of strength and elongation at elevated temperatures.

In the 6th paragraph of **Section 2.4**, we have added that: **Moreover, strength and elongation are explicitly correlated with TGMs intensities, through the linear relationship revealed by high-throughput simulations. By tuning printing parameters and heat treatment durations, the TGMs are tailored to achieve tunable mechanical properties across a wide range.** Besides, in the Supplementary Materials, we have established an inverse-tailoring approach in **Fig. S19**:

Fig. S19 Variations in strength and elongation along the trajectory of heat treatment in Fig.5 f: (a-b) under RT and 980 °C, respectively. The selected cases are representative, as they correspond to the lowest and highest investigated temperatures in this work. The scatters at varying heat treatment durations are extracted from the mapping between intensities of TGMs and mechanical properties in **Fig.5 f**, with the curves fitted based on the power law to explicitly express the strength and elongation using heat treatment durations. Based on the explicit correlations in **a** and **b**, the inverse-tailoring approach has been established, in which the heat treatment duration can be selected for the expected combination of strength and elongation at both RT and 980 °C.

Then, we have added relevant explanations in the 5th paragraph of **Section 2.4**: **Based on Fig. 5 f**, an inverse-tailoring approach is established in **Fig. S19**, where the mechanical properties are explicitly correlated with heat treatment durations, facilitating the selection of processing parameters to achieve the expected combination of strength and elongation. Besides, we also point out the limitations and outlooks for the inverse-tailoring approach: **In future work, more flexible heat treatment strategies can be implemented to realize**

customized combinations of strength and elongation within the microstructure-property maps established in this study.

Additionally, in **Fig. 5 b2**, we have added the validation on the proposed mechanism under the elevated temperature of 850 °C. The relevant explanation has been provided in the 1st paragraph of **Section 2.4**: ‘As shown in **Fig. 5 b**, the enhanced out-of-phase relationship between initial dislocations and slip resistance of type #3 TGMs, is capable of simultaneously enhancing strength and elongation at a higher scanning speed **under both RT and elevated temperature. Additionally, the magnitude of improved elongation is larger at elevated temperatures, aligning well with the trend shown in Fig. 2 b.**’.

Fig. R5 Revised Fig. 5 b2 in manuscript.

References in this comment:

[10] Gao, S. et al. Additive manufacturing of alloys with programmable microstructure and properties. Nature Communications 14, 6752 (2023).

Comment 9: The authors state, "Our work offers a new pathway to simultaneously enhance strength and ductility through AM...." However, the study focuses on a specific Ni-based SX alloy. A major concern remains: can the TGM tailoring approach be generalized to other material systems?

Response: Thanks for the valuable comment. As demonstrated in **Section 2.3**, the TGM mechanism relies on the presence of initial density-graded dislocations surrounding columnar dendrites, which are imparted by the AM process. Regarding the other kinds of AM alloys, the columnar dendrite structures from the literature are shown in **Fig. R6**. Driven by the high cooling rate and large temperature gradient during the AM process, columnar dendritic structures accompanied by solute segregation have been observed within interiors of

columnar grains. These structures are also associated with density-graded dislocations surrounding the columnar dendrites, commonly referred to as dislocation cellular structures formed during rapid solidification^{29,52}. These features resemble the type #1 and #4 TGMs in AM SXs, and the gradient in precipitate size (type #2 and #3) could also occur in some AM alloys^{53,54}. Hence, this study deepens the understanding of the effects of gradient microstructures surrounding columnar dendrites in AM alloys, and provides guidance for tailoring mechanical properties. In future work, the applicability of the TGM mechanism to other AM alloys should be further investigated.

[Figure Redacted]

[Figure Redacted]

[Figure Redacted]

Fig. R6 Columnar dendrite structures in other AM alloys from literature: (a) Solute segregation surrounding dendrites in AM Ni-based Hastelloy C276 polycrystal superalloy²⁴.

(b) AM β -titanium alloys. (c) AM 316L stainless steel⁵⁰. (d) AM Al-Cu alloy³⁵. These figures are reproduced with permission from Elsevier. Essentially, the gradient microstructures around columnar dendrites in other AM alloys resemble the features of TGMs in AM SXs.

In the last sentence of **Abstract**, we have rephrased the statement as: **This study deepens the understanding for the influence of gradient microstructures surrounding columnar dendrites in AM alloys**, and provides guidance for tailoring mechanical properties.

In the last paragraph of **2. Results and discussion**, we have added some discussion on applicability and outlooks: **The proposed mechanism holds promise to be applied on other AM alloys with initial density-graded dislocations and gradient microstructures surrounding columnar dendrites, including AM steel⁵⁰, polycrystal superalloys²⁴, Ti⁵¹ and Al³⁵ alloys. Among them, the high cooling rates and large temperature gradients similarly impart density-graded dislocations, solute segregation^{29,52}, and gradient in precipitate size⁵³ around dendrites within grain interiors. Here, the AM SX is employed as a representative model material eliminating the interference from orientation distribution, thereby revealing the TGM's effects for the typical gradient microstructures around columnar dendrites in AM alloys. In our future work, the applicability of the TGM mechanism to other AM alloys will be further examined.**

References in this comment:

- [24] Qiu, Z. et al. Crystallographic texture and multiscale boundaries mediated creep anisotropy in additively manufactured Ni-based Hastelloy C276 superalloy. *Additive Manufacturing* 83, 104069 (2024).
- [29] Guo, B. et al. Segregation-dislocation self-organized structures ductilize a work-hardened medium entropy alloy. *Nature Communications* 16, 1475 (2025).
- [35] Geng, R., Du, J., Wei, Z. & Ma, N. Multiscale modelling of microstructure, micro-segregation, and local mechanical properties of Al-Cu alloys in wire and arc additive manufacturing. *Additive Manufacturing* 36, 101735 (2020).
- [50] Chen, Y. et al. Multibranches of acoustic emission as identifier for deformation mechanisms in additively manufactured 316L stainless steel. *Additive Manufacturing* 78, 103819 (2023).
- [51] Ng, C. H., Bermingham, M. J. & Dargusch, M. S. Eliminating segregation defects during additive manufacturing of high strength β -titanium alloys. *Additive Manufacturing* 39, 101855 (2021).
- [52] Wang, Y. M. et al. Additively manufactured hierarchical stainless steels with high strength and ductility. *Nature Materials* 17, 63-71 (2018).
- [53] Wang, Z. et al. Microstructure evolution and mechanical properties of the wire + arc additive manufacturing Al-Cu alloy. *Additive Manufacturing* 47, 102298 (2021).
- [54] Vivek Kumar Singh et al. Dissolution of the Laves phase and δ -precipitate formation mechanism in additively manufactured Inconel 718 during post printing heat treatments. *Additive Manufacturing* 81, 104021 (2024).

Comment 10: The reviewer has limited knowledge on CPFEA modelling, and thus could not provide a constructive advice on this part.

Response: Thanks for the comment. The CPFE model is widely recognized for its capability to capture strain and damage localization at the microscale^{1,2}, which is employed as a

powerful tool in this work to reveal the deformation behavior influenced by TGMs. In the revised manuscript, additional validation of the CPFEM model has been conducted against the DIC results, focusing on GND density (**Fig. S8**) and lateral strain distribution (**Fig. 6g**).

References in this comment:

- [1] Xian, J.W., Xu, Y.L., Stoyanov, S. et al. The role of microstructure in the thermal fatigue of solder joints. *Nat Commun* 15, 4258 (2024).
- [2] Jie Ren, et al., Strong yet ductile nanolamellar high-entropy alloys by additive manufacturing, *Nature*, 608, 62–68 (2022).

Reviewer #2:

General comments: The submission focuses on generation of trigonometric gradient microstructures in Ni based single crystals fabricated via AM. High throughput simulations and experimental validations were conducted to analyze the mechanical properties of the TGMs and their strength-ductility tradeoff as opposed to what is commonly observed in conventionally heat treated alloys and metallic materials. With the given findings, it is proved that a simultaneous enhancement of strength and ductility can be achieved by taking advantage of the phase relationship between initial density-graded dislocations and TGMs, which helps balancing the stress/strain distribution and lower chances of damage localization within the material (also in agreement with the designed concave slip resistance distribution in the material). The work is interesting and written well. Given the expertise of the reviewer, the major focus in the given comments and evaluation was on mechanical properties.

Response: We sincerely appreciate the reviewer's positive comments and constructive suggestions. In response, we have carefully revised the manuscript by incorporating the recommended changes. Detailed responses to each comment are provided below.

Comment 1: by “post fabrication” do the authors refer to “post treatment” methods of change of material microstructure?

Response: Thanks for the comment. In the original manuscript, the term ‘post-fabrication’ referred to the fact that, the TGMs introduced by the AM technique in a near-net-shape manner, whereas conventional gradient microstructures require extra processing steps to introduce them, such as laser shock peening or electro-deposition^{1,2}. In the **Abstract** and **Section 2.2** of the revised manuscript, the wording of ‘post-fabrication’ has been revised as ‘**post-treatment**’, which is more accurate.

References in this comment:

- [1] Zhou, W., Ren, X., Yang, Y., Tong, Z. & Chen, L. Tensile behavior of nickel with gradient microstructure produced by laser shock peening. *Materials Science and Engineering: A* 771, 138603 (2020).
- [2] Lin, Y., Pan, J., Zhou, H. F., Gao, H. J. & Li, Y. Mechanical properties and optimal grain size distribution profile of gradient grained nickel. *Acta Materialia* 153, 279-289 (2018).

Comment 2: The reported properties and trends seem interesting and promising. However, can the authors comment on the long-term stability of this? Any potential aging in these architected materials? Especially potential degradation in case of prolonged exposure to elevated temperatures. Please comment on this in the manuscript.

Response: Thanks for the constructive comment. During the revision, we conducted the thermal stability tests on the AM SXs with TGMs. The experimental results indicate that the TGMs in AM SXs have good thermal stability after heat exposure at 900 °C for 10 hours, which is a typical in-service temperature for Ni-based superalloys¹¹. The applied 900 °C is insufficient to drive the diffusion of segregated solutes, as the diffusion is a thermally activated process. Hence, the AM SXs with TGMs show promising thermal stability subjected to long-term heat exposure. In **Fig. S5** within Supplementary Materials, we have added that:

Fig. S5 Thermal stability test results for TGMs in AM SXs: (a)-(b) Distributions of solid-solution elements Cr and Al after heat exposure for 10 hours under 700 °C and 900 °C, respectively. By comparing with the solute segregations in the as-printed state (**Fig. 1 d**), the 700 °C and 900 °C heat exposure has minimal effects on the degree of segregation. The high temperatures ranging from 700 °C and 900 °C are insufficient to drive the diffusion of segregated solutes, as the diffusion is a thermally activated process. Furthermore, the above heat exposure is also unlikely to affect the initial density-graded dislocations, since the solute segregation is highly correlated with initial localized dislocations in AM alloys³. Therefore, the TGMs formed during the AM process show promising thermal stability after heat exposure

from 700 °C to 900 °C for 10 hours, which are typical in-service temperatures of Ni-based superalloys¹¹.

Accordingly, in the last paragraph of **Section 2.1**, we have added that: **As demonstrated in Fig. S5, the AM-induced TGMs exhibit good thermal stability after 10-hour heat exposure at 700 °C and 900 °C.**

References in this comment:

[3] Guo, B. et al. Segregation-dislocation self-organized structures ductilize a work-hardened medium entropy alloy. *Nature Communications* 16, 1475 (2025).

[11] Reed, R. C. *The superalloys: fundamentals and applications*. (Cambridge university press, 2008).

Comment 3: How can the extent of this improvement in ductility and strength impact other critical properties of the alloys? Especially in case of cyclic loading where any change in material, especially slip system can dramatically impact the life time of the component, can what is seen here as improvement show cause even a higher improvement in fatigue performance? Or the opposite, can introduction of the TGMs negatively impact the cyclic plasticity and formation of persistent slip bands? An elaboration on this in the manuscript would be beneficial.

Response: Thanks for the insightful comment. We conducted additional low-cycle fatigue (LCF) tests on AM SXs with TGMs, obtaining five effective fatigue life data. Compared to Ni-based alloys fabricated using conventional methods, the AM SXs exhibit LCF life that are not significantly inferior. Moreover, compared with AM Ni-based polycrystal alloys, the AM SXs exhibit better fatigue performance. Therefore, the introduction of TGMs in AM SXs does not necessarily result in a pronounced degradation of fatigue performance. Although the TGMs enhance the interaction between dendrites and slip bands, the slip band formation is not unique to AM SXs with TGMs, and similarly plays a dominant role in the fatigue failure for other kinds of Ni-based alloys^{17,18}.

Accordingly, in the 2nd paragraph of **Section 2.2**, we have added that: **Additionally, as demonstrated in Fig. S11, the introduction of TGMs does not necessarily cause significant degradation of other mechanical properties, such as low-cycle fatigue performance.**

In **Fig. S11** of the revised Supplementary Materials, we have added that:

$\Delta\varepsilon_t/2$ (%)	LCF lifespan / cycles
0.31	1.72×10^4
0.4	1.22×10^4
0.44	1.12×10^4
0.38	1.06×10^4
0.46	4.67×10^3

Fig. S11 Low-cycle fatigue lifespans of AM SX with TGMs: (a) The fatigue lifespans of AM SX are compared with other Ni-based alloys with similar chemical compositions, including AM Inconel 718¹⁴, as-cast SX/directionally solidified alloys¹⁵, and wrought Inconel 718¹⁵. Compared with AM polycrystal Inconel 718, the AM SXs exhibit significantly improved fatigue performance. Furthermore, since fatigue life is more sensitive to AM-induced defects under low strain amplitude¹⁶, both as-cast SX and wrought Inconel 718 demonstrate notably longer fatigue lifespans than AM SXs at low strain amplitudes. Nevertheless, when the half strain amplitude approximately exceeds 0.5%, the AM SX could achieve fatigue performance comparable to that of the as-cast SX and wrought Inconel 718. Therefore, the introduction of TGMs in the AM SX does not necessarily result in a pronounced degradation of fatigue performance. Although the TGMs enhance the interaction between dendrites and slip bands,

the slip band formation is not unique to AM SXs with TGMs, and similarly plays a dominant role in the fatigue failure of other kinds of Ni-based alloys^{17,18}. **(b)** The low-cycle fatigue tests for AM SX are conducted on a servo-hydraulic fatigue testing system (Instron 8801) using the strain-controlled mode, under a loading frequency of 2 Hz. **(c)** Raw data of fatigue lifespans and associated strain amplitudes from the LCF tests.

References in this comment:

- [14] Gribbin, S., Bicknell, J., Jorgensen, L., Tsukrov, I. & Knezevic, M. Low cycle fatigue behavior of direct metal laser sintered Inconel alloy 718. *International Journal of Fatigue* 93, 156-167 (2016).
- [15] Huichen Yu, X. W. *Materials Data Manual in Aircraft Engine Design* (4th Edition). (Chinese Aviation Industry Press 2010).
- [16] Qu, Z. et al. High fatigue resistance in a titanium alloy via near-void-free 3D printing. *Nature* 626, 999-1004 (2024).
- [17] Stinville, J. C. et al. On the origins of fatigue strength in crystalline metallic materials. *Science* 377, 1065-1071 (2022).
- [18] Sidharth, R., Stinville, J. C. & Sehitoglu, H. Fatigue and fracture of shape memory alloys in the nanoscale: An in-situ TEM study. *Scripta Materialia* 234, 115577 (2023).

Comment 4: Majority of the scientific discussions are relying on the two scale DIC analyses. The authors are asked to provide fractography of the tested specimens, at least on selected cases, in the text. Could you observe any traces of the 111 slip planes on the fracture surface? How did this vary by an increase in the temperature?

Response: Thanks for the valuable comment. In the revised version, we have provided the fractography analysis on the tested specimens, including the cases with and without slip bands under both room temperature and elevated temperature of 980°C, respectively. The selected temperatures for fractography analysis are representative, as they are the lowest and highest testing temperatures in this work. At room temperature, the fracture is dominated by slip and damage localization within slip bands, with dense {111} slip traces observed near the fracture surface. This SEM observation is consistent with DIC results in **Fig. 3 a** where the strain is highly localized within slip bands. On the other hand, at the elevated temperature of 980 °C, microcracks initiate at the inter-dendrites with carbides, which is consistent with the strain localization behavior surrounding carbides captured in high-temperature DIC (**Fig. 3 c**). No slip traces are observed at 980 °C, as the applied stress is insufficient to activate γ' phase shearing, which primarily governs the formation of slip bands¹⁹. As a result, the fractography analysis shows good agreement with DIC results.

Therefore, as the temperature increases, the failure mode changes from shear-dominated (or quasi-cleavage) fracture (induced by strain and damage localization within slip traces) to dimple-dominated ductile fracture (induced by cracking around carbides at inter-dendrites),

which are consistent with the observed strain localization at slip bands (room temperature, **Fig. 3 a**) and carbides (980 °C, **Fig. 3 c**) in DIC analysis.

In **Fig. S14** of **Supplementary Materials**, we have added that:

Fig. S14 Fractography analysis for the tested specimens at room temperature and 980°C: (a1, b1) Height maps for the fracture surface, using the 3D optical microscope (RX-2000, HiROX, Japan). **(a2, b2)** Macroscopic SEM images for the specimens fractured during the *in-situ* tensile tests. **(a3, b3)** Microstructures near the fracture surfaces. Based on the above observations, specimens with evident slip bands show shear-dominated fracture along the {111} slip planes, with dense slip traces observed near the fracture surface. This SEM observation aligns with the strain localization behavior within slip bands captured by DIC (**Fig. 3 a**). In these cases, failure is driven by slip and damage localization within the slip bands. On the other hand, specimens without slip bands display a more ductile fracture mode, characterized by a high density of dimples on the fracture surface. No slip traces are observed at 980 °C, as the applied stress is insufficient to activate γ' phase shearing, which primarily governs the formation of slip traces¹⁹. In the case without slip traces, microcracks

initiate around carbides at the inter-dendrite region due to the strain localization captured in DIC (**Fig. 3 c**). Therefore, the fractography analysis shows good agreement with DIC results. Moreover, these observed damage features can be well captured in our CPFEM modeling: the simulated damage is localized within slip bands (**Fig. 4 b2**) and inter-dendrites with carbides (**Fig. S11 d**), regarding the cases with and without slip bands, respectively. Hence, the simulated cracking sites with damage localization are in good agreement with the experimental observations at both room and elevated temperatures.

Accordingly, in the 1st paragraph of **Section 2.2**, we have added that: **After fracture, dense {111} slip bands with strain localization can be observed in the vicinity of the main crack. (Fig. S14)**. In the 7th paragraph of **Section 3.4**, we have added that: **Firstly, the failure modes for the cases with and without slip bands can be well reproduced in simulations (Fig. S14)**.

References in this comment:

- [19] Guo, Z. et al. Slip Band Evolution Behavior near Circular Hole on Single Crystal Superalloy: Experiment and Simulation. *International Journal of Plasticity* 165, 103600 (2023).

Reviewer #3:

General comments: This manuscript presents a very interesting and comprehensive study on additively manufactured (AM) single crystals, highlighting the significance of multiple trigonometric gradient microstructures (TGMs) surrounding dendrites. Unlike the conventional geometrically necessary dislocations (GNDs) mechanism, the AM-induced TGMs overcome the strength-ductility trade-off through a unique 'phase relationship'. The authors further demonstrate how to effectively tailor mechanical properties by tuning TGMs. This work is both impressive and highly promising, with the potential to attract widespread attention in Nature Communications. However, in prior to the decision, the following comments should be addressed carefully.

Response: We sincerely thank the reviewer for the encouraging remarks and insightful feedback. We are pleased that the significance and potential impact of our work were recognized. In response to the reviewer's valuable suggestions, we have thoroughly revised the manuscript and addressed all comments point by point, as detailed below.

Major Comment 2: Since GND-induced extra strengthening is referred to as a conventional mechanism, it would be beneficial to calculate the GND density from DIC analysis in Fig. 3a–b. This data could then be used to validate the simulated GND density presented in Fig. 4d.

Response: Thanks for the constructive comment. In **Fig. S8** of Supplementary Materials, we have provided the experimental GND density, which shows a similar pattern and magnitude to the simulated GND density, thereby effectively validating our CPFE model.

Fig. S8 Scalar GND density distribution calculated using DIC. The DIC-based GND density is calculated based on the algorithm in literature¹³. The measurement is performed

at 900 °C under 5% global strain, resembling the same loading condition of CPFE-based GND in Fig. 4 d1. DIC-based GND density is primarily concentrated along the boundaries of each slip band, with the dendrite core exhibiting higher GND densities than inter-dendrites. Both spatial distribution and magnitude of the DIC-derived GND density closely match those obtained from CPFE simulations in Fig. 4 d1, providing further validation of the simulated results.

Accordingly, in the last paragraph of Section 3.4, we have added that: The simulated GND density aligns well with the experiment in both pattern and magnitude (Fig. S8).

References in this comment:

[13] Wan, W., et al., Microstructurally-sensitive fatigue crack nucleation in a Zircaloy-4 alloy. Journal of the Mechanics and Physics of Solids 180, 105417 (2023).

Major Comment 3: In Fig. 2b, the data on conventional gradient microstructures in as-cast polycrystals is limited. The authors should provide additional data from the literature to better support the advancements of TGMs.

Response: Thanks for the valuable comment. In the revised Fig. 2 b, we have provided 6 additional strength and elongation data from the literature [39,42]. The amount of data from literature used for comparison increases from 10 to 16. This revision further reinforces the insight that conventional gradient microstructures improve strength at the expense of ductility, therefore highlighting the advances of TGMs. The revised Fig. 2 b is shown as follows:

Fig. R7 Revised Fig 2 b in manuscript.

References in this comment:

[39] Yang, L. et al. Improvement of strength and ductility in a gradient structured Ni fabricated by severe torsion deformation. *Materials Science and Engineering: A* 826, 141980 (2021).
 [42] Fu, W., Huang, Y., Sun, J. & Ngan, A. H. W. Strengthening CrFeCoNiMn0.75Cu0.25 high entropy alloy via laser shock peening. *International Journal of Plasticity* 154, 103296 (2022).

Major Comment 4: In Fig. 3e–f, it would be helpful to plot the strain distributions for the samples without TGMs along the B–B’ and A–A’ paths. This would more explicitly highlight the effects of TGMs on the partitioning of normal strain and lateral strain.

Response: Thanks for the helpful comment. In the revised Fig. 3 f, we have added the strain distribution in the case without TGMs shown in Fig. 3 d. The strain partitioning disappears after the TGMs are removed, which experimentally highlights their impact on deformation.

Fig. R8 Revised Fig 3 f in manuscript.

Due to the absence of slip traces in the *in-situ* tests without TGMs, the path A-A’ within slip traces cannot be identified. From the normal strain pattern shown in Fig. 3 c, after removing TGMs, there is no interaction between slip traces and dendrites, and the normal strain is localized at carbides instead. Hence, the case without TGMs exhibits a normal strain pattern that is distinct from the one with TGMs, further highlighting the impact of TGMs on the deformation behavior.

In the 3rd paragraph of Section 2.2, we have added that: **After removing TGMs through heat treatment (Fig. S17), there is no strain partitioning and interaction between slip traces and**

dendrites in the absence of TGM, and the ε_{xx} tends to be localized around carbides instead (Fig. 3 c). Hence, the case without TGMs exhibits a strain pattern distinct from the one with TGMs, highlighting the impact of TGMs on the deformation behavior.

Major Comment 5: Regarding the validation of the proposed CPFE model in Fig. 6g, only a single global strain level is used to validate the simulation results for lateral strains. It is recommended to incorporate additional global strain levels to further validate or calibrate the model.

Response: Thanks for the constructive comment. An additional comparison between the simulated and experimental lateral strain has been added in the revised manuscript to further validate the CPFE model. Both simulation and experiment indicate that, the dendrite core shows a higher absolute value of ε_{yy} , with more pronounced partitioning at the early stage of deformation.

Fig. R9 Revised Fig. 6 g in manuscript.

In the last paragraph of **3.4 Crystal plasticity modelling**, we have added that: **Moreover, the dendrite core shows a higher absolute value of ε_{yy} (Fig. 6 d2), with more pronounced partitioning at the early stage of deformation (Fig. 6 g).** In CPFE simulations, the evolution of both normal and lateral strain distributions aligns well with the DIC results.

Minor Comment 1: Eq. (5) is the definition formula of GND. The authors should further clarify how to implement GND in CPFE simulations.

Response: Thanks for the valuable comment. In **3.4 Crystal plasticity modelling**, below Eq. (5), we have further explained that: **where the GNDs contribute to the total dislocation**

hardening through Eq. (S.5). When implementing Eq. (5), the slip system-dependent accumulated slip γ^s is calculated by integrating the slip increments provided by Eq. (3). Subsequently, the edge and screw GND densities are resolved for each slip system based on the gradients of slip in Eq. (5). The presented scalar GND density corresponds to the second norm of the slip system–resolved GND densities.

Minor Comment 2: In Fig 6 a, please provide the definition of $F_{p,0xx}$ in the caption.

Response: Thanks for the careful review. In the caption of **Fig. 6**, we have further illustrated that: **(a2)** Directions of residual deformation and residual stresses, where $F_{p,0}^{xx}$ is the dominant xx component within the residual deformation gradient tensor $\mathbf{F}_{p,0}$, which is used to capture the residual stresses, with the detailed demonstration in **Fig. S6** and **Supplementary Material S1**.